



# Multi-model assessment of climatologies in the upper troposphere–lower stratosphere using the IAGOS data

Yann Cohen[1], Didier Hauglustaine[1], Nicolas Bellouin[2], Marianne Tronstad Lund[3], Sigrun Matthes[4], Agnieszka Skowron[5], Robin Thor[4], Ulrich Bundke[6], Andreas Petzold[6,7], Susanne Rohs[6], Valérie Thouret[8], Andreas Zahn[9], and Helmut Ziereis[4]

[1]Laboratoire des Sciences du Climat et de l'Environnement, LSCE-IPSL (CEA-CNRS-UVSQ), Université Paris-Saclay, 91191 Gif-sur-Yvette, France

[2]Department of Meteorology, University of Reading, Reading, UK

[3]CICERO Center for International Climate Research, Oslo, Norway

[4]Deutsches Zentrum für Luft und Raumfahrt, Institut für Physik der Atmosphäre, Oberpfaffenhofen, Germany

[5]Faculty of Science and Engineering, Manchester Metropolitan University, Manchester, United Kingdom

[6]Forschungszentrum Jülich GmbH, Institute of Energy and Climate Research 8 - Troposphere, Jülich, Germany

[7]Institute for Atmospheric and Environmental Research, University of Wuppertal, Wuppertal, Germany

[8]Laboratoire d'Aérologie, Université de Toulouse, CNRS, UPS, France

[9]Institute of Meteorology and Climate Research, Karlsruhe Institute of Technology, Karlsruhe, Germany

**Correspondence:** yann.cohen.09@gmail.com

**Abstract.**

The evaluation of global chemistry–climate/transport models in the upper troposphere–lower stratosphere (UTLS) is a important step towards a better understanding of the chemical composition near the tropopause, and therefore towards a more accurate assessment of the impact of $NO_x$ emissions in this region of the atmosphere, notably by subsonic aviation. For this

purpose, the current study focuses on an evaluation of long-term simulations from five global models based on in-situ measurements on board passenger aircraft (IAGOS). Most simulations span over the 1995–2017 time period, and follow a common protocol among the models. The assessment focuses on climatological averages of ozone ($O_3$), water vapour ($H_2O$), carbon monoxide (CO), and reactive nitrogen compounds ($NO_y$). In the extra-tropics, the models reproduce the seasonality of ozone, water vapour and $NO_y$ in both the upper troposphere (UT) and the lowermost stratosphere (LS), but none of them reproduces

the CO springtime maximum in the UT. The tropospheric tracers (CO and $H_2O$) tend to be underestimated by the models,



which is consistent with an overestimation of the cross-tropopause exchange, but does not exclude other factors as an underestimation of CO emissions, an underestimation of transport from the surface, or an overestimated CO oxidation by the hydroxyl radical (OH). Ozone is systematically overestimated in the UT by most models, and the $NO_x$ background appears as the main contributor to the ozone variability across the models. The partitioning between $NO_y$ species changes drastically

across the models, and acts as a source of uncertainty on the $NO_x$ mixing ratio and on its impacts on atmospheric composition and particularly on the response to aviation $NO_x$ emissions. However, independently on the mean biases, we highlight some well-reproduced geographical and seasonal distributions, as the ITCZ seasonal shifts above Africa, the upper-tropospheric $H_2O$ maximum in the Asian summer monsoon, and the extratropical ozone ($H_2O$) in the LS (UT) that show a high correlation with the observations. These features are encouraging regarding the simulated dynamics in both the troposphere and the strato-

sphere. The current study confirms the importance of an accurate separation between the UT and LS using a dynamical tracer for model results evaluation but also for model intercomparisons.

## 1   Introduction

The upper troposphere–lower stratosphere (UTLS) is a complex transition region between the troposphere and the stratosphere. Its dynamical structure limits the exchanges of air between the two layers, thus playing an important role in their respective

quantities of short-lived tracers as ozone ($O_3$), water vapour ($H_2O$), carbon monoxide (CO) and nitrogen oxides ($NO_x$) that are classified as essential climate variables (Bojinski et al., 2014). The UTLS is also a key region regarding radiative forcing, as its colder temperatures maximize the difference between absorbed and emitted long-wave radiation by several radiatively active species (e.g. Lacis et al., 1990). Changes in greenhouse gases concentrations like ozone and water vapour thus have more impact on surface temperature when they are located at these altitudes (e.g. Iglesias-Suarez et al., 2018; Riese et al.,

2012; de F. Forster and Shine, 1997).

$NO_x$ is a necessary ingredient for the ozone formation in the troposphere. Its emissions mostly take place at the surface, yet the short $NO_x$ lifetime in the boundary layer limits considerably its transport into the upper troposphere (UT). In contrast, the $NO_x$ injected directly into these altitudes contribute significantly to the $NO_x$ mixing ratios, first by lightning (Allen et al., 2010; Cooper et al., 2009) then by aviation, which traffic increased since the 1950s (Lee et al., 2021). Due to the lesser $NO_x$

background favouring the $NO_x$-limited regime, the $NO_x$ emitted in the upper troposphere is more efficient in producing ozone than in the boundary layer (e.g. Nussbaumer et al., 2023; Hoor et al., 2009, for lightning and aviation respectively). Combined with the presence of water vapour, the ozone production enhances the concentrations of hydroxyl radical (OH) which acts as a





sink of methane and CO, but also converts the background sulfur dioxide ($SO_2$) into sulfate ($SO_4$), thus enhancing the aerosol production in the UTLS (e.g. Joppe et al., 2024).

Modelling accurately the UTLS behaviour is thus an important step towards a better representation of the impact of the aviation $NO_x$ emissions. The ozone production due to aircraft emissions depends on the background chemical composition, notably in $NO_x$, ozone and CO (Hegglin et al., 2006). Also, nitric acid ($HNO_3$) is soluble and can be washed out rapidly in moist conditions, which acts as a sink of $NO_y$ species. As $NO_x$ are converted back and forth into other $NO_y$ species (mostly $HNO_3$ at these altitudes, but also PAN, provided that peroxyacetyl radicals are present), the $HNO_3$ scavenging is a sink for

$NO_x$ as well. The $NO_y$ lifetime thus depends on the model parameterization of precipitation, but also on the OH quantities that convert $NO_x$ into $HNO_3$, thus making $NO_y$ more vulnerable to scavenging. The tropopause height and the cross-tropopause exchanges can be critical parameters as well. And as each global chemistry-climate model (CCM) and chemistry-transport model (CTM) has its own chemical scheme, as well as its own convection parameterization, the uncertainties in the chemical background partly arises from the intermodel variability. For example, according to the results from the ACCMIP (Atmospheric

Chemistry and Climate Model Intercomparison Project: Lamarque et al., 2013) modeling experiment, Finney et al. (2016) found a $6.5 \pm 4.7$ ratio in ozone production efficiency between lightning $NO_x$ ($LNO_x$) and surface $NO_x$, the high variability originating mainly from the altitude of the $LNO_x$ emissions and on the treatment of volatile organic compounds (VOCs) in the models.

        Assessing the models abilities in reproducing the climatological chemical background in the UTLS provides a degree of

confidence in the model results, helps to understand the sensitivity of the models responses to the aircraft emissions in the background conditions, but can also help to identify the modeled physical and chemical processes which representation needs to be improved. Since the UTLS is not a homogeneous layer, the assessment in this area takes benefit from a separation of the air masses into several categories treated separately. In the extratropics, the UTLS can be divided into an upper troposphere (UT), a transition zone enveloping the tropopause, and a lowermost stratosphere (LMS, or LS as noted hereafter). Ozone and $NO_y$

are present in larger amounts in the lowermost stratosphere while CO and water vapour are abundant in the upper troposphere (e.g. Cohen et al., 2018; Stratmann et al., 2016; Petzold et al., 2020; Zahn et al., 2014). Their comparisons with observations in the different layers can thus be used to assess the stratosphere–troposphere exchanges. More precisely in the LS, ozone and $NO_y$ can be used to assess the models ability in reproducing the effects of stratospheric processes as the Brewer–Dobson circulation. CO is mostly emitted by combustion processes, such as biomass burning and surface anthropogenic emissions

(aviation being a low CO emitter), and can thus be used to assess the surface emissions, convection and the troposphere-to-



stratosphere transport. Last, $NO_y$ is emitted by combustion processes and by lightning, thus it can also be used to identify aviation emissions, lightning emissions, or surface emissions uplifted in ascending motions.

A wide variety of observation data sets are available and commonly used in model assessments. Satellite measurements regularly cover a large area, but their vertical resolution is too coarse to characterize a region as thin as the UTLS. On the contrary, ozonesondes (Hodnebrog et al., 2011) and Lidar (Light Detection and Ranging) provide regular and accurate vertical profiles, but limited to the vicinity of the ground stations. Airborne campaigns sample the atmospheric composition up to 16 km above sea level, and their merged climatologies (Tilmes et al., 2010) have been used in several multimodel assessments (e.g. Hegglin et al., 2010; Gettelman et al., 2010), but the data are sparse in space and time which limits the representativeness of the measurement climatologies. In the framework of the IAGOS (In-service Aircraft for a Global Observing System: Petzold et al., 2015) research infrastructure, regular in situ measurements on board several commercial aircraft provide accurate information in the extratropical UTLS and in the tropical UT, though the very top of this layer is higher than the cruise altitudes. The monitoring began in 1994 for ozone and $H_2O$, 1997 for $NO_y$ and 2001 for CO, with an abundant sampling in most of the northern extratropics and in several tropical transects. The IAGOS database fits particularly well with the ACACIA project since the spatial distribution of the measurements coincide with the aircraft traffic, thus providing essential information on a region where aviation emissions have their strongest impact (Hoor et al., 2009; Hodnebrog et al., 2011, 2012; Søvde et al., 2014; Terrenoire et al., 2022).

The IAGOS database has already been involved in model assessments, but either on a short time period (Law et al., 2000; Brunner et al., 2003), on a restricted area (Gaudel et al., 2015; Tilmes et al., 2016; Young et al., 2018; David et al., 2019), and/or without a IAGOS mask applied to the model output. For this purpose, the Interpol-IAGOS software (Cohen et al., 2021) projects the whole IAGOS data set onto the model grid, and applies a mask on the non-sampled grid cells. As a first application, it has been used to assess (bi-)decadal climatologies in ozone and CO for the MOCAGE (Josse et al., 2004; Guth et al., 2016) model (Cohen et al., 2021) with monthly output for the Chemistry-Climate Model Initiative (CCMI: Eyring et al., 2013), and to assess climatologies in ozone, water vapour, CO, and $NO_y$ for the LMDZ–INCA model with daily output (Cohen et al., 2023), and has proven useful to highlight some model skills as well as biases, either in the UT and LS separately or in the whole UTLS without airmass distinction.

The current study aims to extend the former assessment to the climatologies from long-term simulations from five state-of-the-art chemistry–climate or chemistry–transport models (CCMs or CTMs), involved in a multimodel experiment in the framework of the ACACIA (Advancing the Science for Aviation and Climate) European Union project. This experiment focuses on the present-day (2014–2018) and future (2050) impact of aircraft $NO_x$ and aerosol emissions on the atmospheric

composition and therefore on climate, and consists of the analysis of runs with and without aircraft emissions. While a companion paper focuses on the present-day sensitivity of the modelled atmospheric composition to aviation emissions (Cohen et al., in prep.), the current paper consists of assessing (bi-)decadal climatologies derived from the main run of every model against the IAGOS data, in the UT and in the LS separately, as done in Cohen et al. (2023) for the LMDZ–INCA model.

In this paper, Section 2 describes the participating models and their output, the common simulation setup, the IAGOS observations and their use in the assessment of the model climatologies. The results are shown in Section 3, starting with an overview of the models biases in the whole area covered by IAGOS (Section 3.1), followed by the analysis of the models skills in the extratropical UTLS (Section 3.2) then in the tropical UT (Section 3.3). The conclusion of this analysis is provided in Section 4.

## 2  Materials and methodology

This section presents the tools involved in this study. The first part is dedicated to the description of the setup for the standard runs that are compared to the observations. The second part describes the participating models. The third subsection describes the observation data set and the method used in the assessment of the models.

In this experiment, each model output is projected onto a common grid, with a horizontal resolution of $1.875°$ E x $1.25°$ N combining the most resolved longitude and latitude coordinates separately among the models, and a vertical resolution of 20 hPa at the cruise altitudes. As each individual model output, the common grid has a daily resolution. Except for MOZART3, each model also provided an Ertel potential vorticity field (PV) for the separation between UT and LS.

### 2.1  Simulation setup

The historical global anthropogenic emissions are taken from the Community Emissions Data System inventories (CEDS: Hoesly et al., 2018), and the historical biomass burning emissions from the BB4CMIP inventory (biomass burning emissions for CMIP6: van Marle et al., 2017). The emissions after 2014 are taken from the SSP3-7.0 scenario (Gidden et al., 2019). The aircraft emissions are taken from the anthropogenic emissions inventories as well (both historical and future scenarios), after applying the corrections presented in Thor et al. (2023). The historical runs generally cover the period 1994–2017 (2001–2017 for OsloCTM3 and 1997–2007 for MOZART3), providing robust climatologies that are compared with aircraft observations over the same period. The runs are nudged or forced by horizontal winds taken from a reanalysis.



**Table 1.** Description of the participating models. The acronyms and abbreviations are explicited here. In the first column, the abbreviations horiz., vert., hom., phot., and het. have the respective meanings: horizontal, vertical, homogeneous, photolytic, and heterogeneous. Among the aerosol categories, $SO_4$, $NO_3$, BC, OC, and OM represent respectively sulfate, nitrate, black carbon, organic carbon, and organic matter. In the references, G2001 stems for Grewe et al. (2001), PR92 and PR97 stem for Price and Rind (1992) and Price et al. (1997), O2010 stems for Ott et al. (2010), and P1998 stems for Pickering et al. (1998).

| Model | EMAC | LMDZ-INCA | MOZART3 | OsloCTM3 | UKESM1.1 |
|---|---|---|---|---|---|
| Operated | DLR | LSCE (IPSL) | MMU | CICERO | UREAD |
| Model type | CCM (CTM mode) | CCM (CTM mode) | CTM | CTM | CCM (nudged) |
| Reanalysis | ERA-Interim | ERA-Interim | ERA-Interim | ECMWF OpenIFS | ERA5 |
| GCM | ECHAM5 | LMDZ | – | – | UM |
| Horiz. resolution | 2.8°E x 2.8°N | 2.5°E x 1.3°N | 2.8°E x 2.8°N | 2.25°E x 2.25°N | 1.875°E x 1.25°N |
| Vertical levels | 90 | 39 | 60 | 60 | 85 |
| Vert. resolution (hPa) near cruise levels | 15–20 | 25–40 | 20–25 | 25–30 | 15–20 |
| Top level (hPa) | 0.010 | 0.012 (80 km) | 0.10 | 0.10 | 0.002 |
| Chemistry | | | | | |
| Total species | 160 | 123 | 108 | 190 | 81 |
| Aerosol species | – | 23 | – | 56 | |
| Hom. reactions | 265 | 234 | 218 | 263 | 224 |
| Phot. reactions | 82 | 43 | 71 | 61 | 59 |
| Het. reactions | 12 | 30 | – | 18 | 5 |
| Aerosol categories | – | $SO_4$, $NO_3$, BC, OC, dust, sea-salt | – | $SO_4$, $NO_3$, BC, OC, dust, sea-salt | $SO_4$, $NO_3$, BC, OM, dust, sea-salt |
| Emissions | | | | | |
| Lightning | G2001 | PR92; O2010 | PR97; P1998 | PR92; O2010 | PR92 (calibrated) |
| Biogenic VOCs | | ORCHIDEE model | POET | MEGAN-MACC | Dedicated scheme |
| Biomass burning | | BB4CMIP | BB4CMIP | BB4CMIP | BB4CMIP |

## 2.2 Participating models

In this section, Table 1 summarizes the key model characteristics, then further detail is given for each model in Sections 2.2.1–2.2.5.

### 2.2.1 EMAC

The ECHAM/MESSy Atmospheric Chemistry (EMAC) model is a numerical chemistry and climate simulation system that includes sub-models describing tropospheric and middle atmosphere processes and their interaction with oceans, land and human influences (Jöckel et al., 2010). It uses the second version of the Modular Earth Submodel System (MESSy2) to link multi-institutional computer codes. The core atmospheric model is the 5th generation European Centre Hamburg general circulation model (ECHAM5, Roeckner et al., 2006). The physics subroutines of the original ECHAM code have been modularized and reimplemented as MESSy submodels and have continuously been further developed. Only the spectral transform core, the flux-form semi-Lagrangian large scale advection scheme, and the nudging routines for Newtonian relaxation are remaining from





ECHAM. For the present study we applied EMAC (MESSy version 2.55.2) in the T42L90MA-resolution, i.e. with a spherical

truncation of T42 (corresponding to a quadratic Gaussian grid of approximately 2.8 by 2.8 degrees in latitude and longitude)

with 90 vertical hybrid pressure levels up to 0.01 hPa. The usage of a so-called quasi chemistry–transport mode (QCTM,

Deckert et al., 2011) enables binary identical simulations with respect to atmospheric dynamics and perturbations in chemistry

can be detected with a high signal-to-noise ratio. The applied model setup comprised the Module Efficiently Calculating the

Chemistry of the Atmosphere (MECCA) used for tropospheric and stratospheric chemistry calculations with the possibility

of extending to the mesosphere and oceanic chemistry (Sander et al., 2019). Reaction mechanism includes ozone, methane,

$HO_x$, $NO_x$, NMHCs, halogens and sulfur chemistry. Radiative transfer calculations are performed using the submodel RAD

(Dietmüller et al., 2016).

**2.2.2   LMDZ–INCA**

The LMDZ-INCA global chemistry-aerosol-climate model (hereafter referred to as INCA) couples on-line the LMDZ general

circulation model (Laboratoire de Météorologie Dynamique, version 6: Hourdin et al., 2006) and the INCA model (INteraction

with Chemistry and Aerosols, version 5: Hauglustaine et al., 2004). The interaction between the atmosphere and the land

surface is ensured through the coupling of LMDZ with the ORCHIDEE dynamical vegetation model (ORganizing Carbon

and Hydrology In Dynamic Ecosystems, version 9: Krinner et al., 2005). In the present configuration, the model includes 39

hybrid vertical levels extending up to 70 km. The horizontal resolution is 1.25° in latitude and 2.5° in longitude. The primitive

equations in the general circulation model (GCM) are solved with a 3 min time-step, large-scale transport of tracers is carried

out every 15 min, and physical and chemical processes are calculated at a 30 min time interval. For a more detailed description

and an extended evaluation of the GCM, we refer to Hourdin et al. (2006).

INCA initially included a state-of-the-art $CH_4$-$NO_x$-CO-NMHC-$O_3$ tropospheric photochemistry (Hauglustaine et al., 2004;

Folberth et al., 2006). The tropospheric photochemistry and aerosols scheme used in this model version is described through

a total of 123 tracers including 22 tracers to represent aerosols. The model includes 234 homogeneous chemical reactions, 43

photolytic reactions and 30 heterogeneous reactions. The gas-phase version of the model has been extensively compared to

observations in the lower troposphere and in the upper troposphere. For aerosols, the INCA model simulates the distribution

of aerosols with anthropogenic sources such as sulfates, nitrates, black carbon, particulate organic matter, as well as natural

aerosols such as sea salt and dust. Ammonia and nitrate aerosols are considered as described by Hauglustaine et al. (2014).

The model has been extended to include an interactive chemistry in the stratosphere and mesosphere. Chemical species and

reactions specific to the middle atmosphere were added to the model. A total of 31 species were added to the standard chemical

scheme, mostly belonging to the chlorine and bromine chemistry, and 66 gas-phase reactions and 26 photolytic reactions
(Terrenoire et al., 2022; Pletzer et al., 2022).

In this study, meteorological data from the European Center for Medium-Range Weather Forecasts (ECMWF) ERA-Interim reanalysis have been used to constrain the GCM meteorology and allow a comparison with measurements. The relaxation of the GCM winds towards ECMWF meteorology is performed by applying at each time step a correction term to the GCM zonal and meridional wind components with a relaxation time of 3.6 h. The ECMWF fields are provided every 6 hours and
interpolated onto the LMDZ grid.

The ORCHIDEE vegetation model has been used to calculate off-line the biogenic surface fluxes of isoprene, terpenes, acetone and methanol as well as NO soil emissions as described by Messina et al. (2016). The lightning $NO_x$ parameterization is described in Jourdain and Hauglustaine (2001). The lightning frequency follows the parameterization from Price and Rind (1992). In this simulation, a rescaling constrains the mean global flash rate at 46.3 flash yr$^{-1}$, consistently with the annual
climatologies derived from both Lightning Imaging Sensor and Optical Transient Detector (LIS–OTD) satellite instruments in Cecil et al. (2014), from 1995 until 2010. This rescaling accounts for the different LIS and OTD sampled latitude bands, and for their different sampling periods. The lightning $NO_x$ ($LNO_x$) emissions are then redistributed vertically, based on Ott et al. (2010).

### 2.2.3 MOZART3

Model for OZone And Related chemical Tracers, version 3 (MOZART3) is an offline, global chemical transport model, extensively evaluated (Kinnison et al., 2007) and used for a range of various applications (Liu et al., 2009; Wuebbles et al., 2011), including studies dealing with the impact of aviation emissions on atmospheric composition (Søvde et al., 2014; Skowron et al., 2015). MOZART3 accounts for advection based on the flux-form semi-Lagrangian scheme (Lin and Rood, 1996), shallow and mid-level convection (Hack, 1994), deep convective routine (Zhang and McFarlane, 1995), boundary layer exchanges (Holtslag
and Boville, 1993), or wet and dry deposition (Brasseur et al., 1998; Müller, 1992).

The model reproduces detailed chemical and physical processes from the troposphere through the stratosphere. The chemical mechanism consists of 108 species, 218 gas-phase reactions, and 71 photolytic reactions including the photochemical reactions associated with organic halogen compounds. The species included within this mechanism are members of the $O_x$, $NO_x$, $HO_x$, $ClO_x$ and $BrO_x$ chemical families, along with $CH_4$ and its degradation products. A non-methane hydrocarbon oxidation scheme
is also represented. The kinetic and photochemical data is based on the NASA/JPL evaluation (Sander et al., 2006).





The horizontal resolution used in this study is T42 (2.8° x 2.8°) and vertically the model domain spans 60 layers between the surface and 0.1 hPa. Vertical resolution is 700–900 m at aircraft cruise altitudes (250–200 hPa). The transport of chemical compounds as well as the hydrological cycle is driven by the meteorological fields from ECMWF Interim 6-h reanalysis (ERA-Interim).

The surface and aviation emissions represent the years 2014–2018. The anthropogenic and biomass burning emissions are taken from CEDS version 2021 and GFEDv4, respectively, while the biogenic emissions are taken from POET (Granier et al., 2005). The parameterization of $NO_x$ emissions from lightning follows the assumption that the lightning frequency depends on the convective cloud top height and the ratio of cloud-to-cloud versus cloud-to-ground lightning depends on the cold cloud thickness (Price et al., 1997). The lightning $NO_x$ emissions are distributed vertically through the convective column according

to observed profiles based on Pickering et al. (1998). The lightning source is scaled to provide a total of 4.7 Tg(N) yr$^{-1}$, with diurnal and seasonal fluctuations based on the model meteorology. The patterns of lighting $NO_x$ distribution in MOZART3 show a general agreement with LIS and OTD climatology datasets (Skowron et al., 2021). Simulations were preceded by a one-year spin-up.

### 2.2.4   OsloCTM3

OsloCTM3 is a global, offline chemical transport model, driven by 3-hourly meteorological forecast data from the European Centre for Medium-Range Weather Forecasts (ECMWF) Integrated Forecast System (IFS) model (Søvde et al., 2012). The default horizontal resolution is 2.25° x 2.25°, with an option to run at 1° x 1°. In the vertical, the model has 60 levels with the uppermost centered at 0.1 hPa. The model code is openly available from github: https://github.com/NordicESMhub/OsloCTM3.

The chemistry of the OsloCTM3 covers both tropospheric and stratospheric chemistry, treated by separate modules (Berntsen
and Isaksen, 1997; Stordal et al., 1985). The tropospheric code is stand-alone, but the stratospheric code needs the tropospheric chemistry module to work. The kinetics are based on JPL 2006 (Sander et al., 2006), while the photodissociation coefficients are calculated on-line using the Fast-JX scheme (Prather, 2009). The numerical integration of chemical kinetics is done applying the Quasi Steady State Approximation (QSSA: Hesstvedt et al., 1978), using three different integration methods depending on the chemical lifetime of the species. The model also treats the main anthropogenic and natural aerosol species (sulfate,
nitrate/ammonium, black carbon, primary and secondary organic aerosol, dust and sea salt). The aerosol schemes are described in more detail in Lund et al. (2018).

The model transport covers large-scale advection treated by the second order moments (SOM) scheme (Prather, 1986), convective transport based on Tiedtke (1989) and boundary layer mixing based on Holtslag et al. (1990). Scavenging covers





dry deposition, i.e. uptake by soil or vegetation at the surface, and washout by convective and large scale rain (Søvde et al.,
215   2012).

For ACACIA, one-year simulations are performed, with 6 months spin-up. Anthropogenic emissions are from CEDS version 2021 with GFEDv4 biomass burning and MEGAN-MACC year 2010 biogenic emissions. Lightning $NO_x$ emissions are calculated online (Søvde et al., 2012), as are dust and sea salt fluxes (Lund et al., 2018, and references therein).

Lightning $NO_x$ emissions are calculated from the convective fluxes provided by the meteorological input data using the
algorithm based on cloud-top height from Price and Rind (1992), with a scaling that matchs lightning flash rates observed by the OTD and LIS. In-cloud flash rate depends on whether the surface is land or ocean. The model distributes $LNO_x$ emissions vertically through the convective column according to observed profiles (Ott et al., 2010) for 4 world regions. These profiles are scaled vertically to match the height of each convective plume in the CTM and already account for vertical mixing of lightning $NO_x$ within the cloud. Geographic region definitions are from Allen et al. (2010) and Murray et al. (2012).

### 2.2.5   UKESM1.1

The U.K. Earth System Model version 1 (UKESM1: Sellar et al., 2019) is a global climate model made by coupling atmosphere, ocean, sea ice, and land surface models. In this study, UKESM1 is used in its atmosphere-only configuration, where ocean sea surface temperature and sea ice distributions are prescribed from previous simulations with the fully coupled model. The atmosphere model is built on the Met Office Unified Model (Walters et al., 2019), decomposing the atmosphere in 85 terrain-
following hybrid vertical levels up to an altitude of 85 km. The horizontal resolution is $1.25° \times 1.875°$.

Atmospheric chemistry is simulated by the stratosphere-troposphere StratTrop chemistry scheme (Archibald et al., 2020) of the UK Chemistry and Aerosols (UKCA) sub-model. StratTrop unifies two originally separate tropospheric and stratospheric chemistry modules, described by O'Connor et al. (2014) and Morgenstern et al. (2009), respectively. StratTrop simulates Ox, HOx and NOx chemistry based on 15 emitted species, including NO, CO, and aerosol precursor gases, and 7 long-lived
species including $CH_4$, which are constrained by surface concentrations. Tracer advection, convective transport and boundary layer mixing are simulated by the Unified Model (Walters et al., 2019). Wet deposition follows Giannakopoulos et al. (1999), while dry deposition depends on the surface types simulated by the land surface model, as described in Archibald et al. (2020). Photolysis rates are computed interactively by the Fast-JX scheme (Neu et al., 2007) depending on three-dimensional radiation. In terms of aerosols, UKCA simulates the mass and number of sulfate, nitrate, black carbon, primary and secondary organic,
mineral dust and sea salt aerosols (Mulcahy et al., 2018).





Lightning $NO_x$ emissions are described in section 2.6.3 of Archibald et al. (2020). They are calculated following Price and Rind (1992), where lightning flash density depends on cloud top height and surface type (land or ocean). The scheme distinguishes the energy discharged by cloud-to-cloud and cloud-to-ground flashes and uses a spatial calibration factor to make the scheme independent of model resolution. The scheme is only applied when cloud depth reaches at least 5 km according to the convection scheme. $NO_x$ emissions are distributed linearly with the logarithm of pressure. They have been calibrated to reach an average global annual emission rate of 5.98 TgN yr$^{-1}$ over the period 2005 to 2014.

In this study, UKESM1 simulated the period 1990–2018, using CMIP6 historical and, from 2015 onwards, SSP3-7.0 emissions as monthly distributions. The CMIP6 aircraft emission inventories were corrected for the mistake identified by Thor et al. (2023). Emissions of sea salt and mineral dust aerosols, and biogenic VOCs are interactive. The model was nudged to 6-hourly horizontal wind speed distributions from ERA5.

## 2.3 The IAGOS data

The IAGOS research infrastructure (http://www.iagos.org, last access: November 2022) provides long-term routine in situ observations of chemical species on board a fleet of several passenger aircraft. Its predecessors MOZAIC (Measurements of water vapor and OZone by Airbus In-service airCraft: Marenco et al., 1998) and CARIBIC (Civil Aircraft for the Regular Investigation Based on an Instrument Container: Brenninkmeijer et al., 1999, 2007; Stratmann et al., 2016) relied on the same principle. The MOZAIC measurements began on board five equipped aircraft measuring ozone and water vapour since August 1994. CO observations began in December 2001 and $NO_y$ measurements were operational on one aircraft between April 2001 and May 2005. CARIBIC samples a wide variety of atmospheric species since 1997 from one single aircraft, including the ones measured by MOZAIC. Since the merge of the two programs in 2008, their respective databases are referred as IAGOS-CORE and IAGOS-CARIBIC. In the present study, we consider them as a single database called IAGOS hereafter, approach validated by Blot et al. (2021) for ozone and CO. The period we are analysing spreads from Aug. 1994 until Dec. 2017.

The main characteristics of the IAGOS instruments relevant for this study are indicated in Table 2. Concerning the IAGOS-CORE instruments, further information is available in Thouret et al. (1998) for ozone, in Nédélec et al. (2003); Nédélec et al. (2015) for CO, in Helten et al. (1998); Neis et al. (2015a, b); Rolf et al. (2023) for water vapour, and in Volz-Thomas et al. (2005); Pätz et al. (2006) for $NO_y$. Note that the IAGOS-CORE water vapour measurements have an accuracy of 6 % RHL (relative humidity with respect to liquid water) in the vicinity of the midlatitude thermal tropopause (Smit et al. (2014); Petzold et al. (2020)). Due to a moist bias in the IAGOS-CORE $H_2O$ observations for the driest air masses (RHL < 5 %) which are encountered in the lower stratosphere, this study does not quantify the model $H_2O$ biases elsewhere than in the upper



troposphere. Concerning the IAGOS-CARIBIC instruments, further information is available in Zahn et al. (2012) for ozone, in
Scharffe et al. (2012) for CO, in Zahn et al. (2014); Dyroff et al. (2015) for water vapour, and in Ziereis et al. (2000); Stratmann
et al. (2016) for $NO_y$. More precisely, the latter has a total measurement uncertainty of 6.5 % (8 %) for a measured mixing
ratio of 1 ppb (0.5 ppb). For both programs, the time response of the water vapour sensors decreases with the measured water
content.

**Table 2.** Characteristics of the IAGOS instruments measuring ozone, CO, water vapour and $NO_y$.

| Observation system | Species | Instrument | Accuracy | Precision | Time response |
|---|---|---|---|---|---|
| IAGOS-CORE | $O_3$ | UV absorption spectrometer | 2 ppb | 2 % | 4 s |
| | CO | IR absorption spectrometer | 5 ppb | 5 % | 30 s |
| | $H_2O$ | Capacitive hygrometer | 5 % RHL | | 5–300 s |
| | $NO_y$ | Chemiluninescence gold converter | 5 ppt | 5 % | 4 s |
| IAGOS-CARIBIC | $O_3$ | Dry chemiluminescence detector & UV absorption spectrometer | 1.5 ppb | 1 % | 0.2 s 4 s |
| | CO | UV resonance fluorescence | < 2 ppb | 1–2 ppb | 2 s |
| | $H_2O$ | Photoacoustic laser spectrometer & frost-point hygrometer | < 1 ppm | < 3 % | 4–20 s 5–90 s |
| | $NO_y$ | Chemiluninescence gold converter | 6.5–8 % | | 1 s |

## 2.4 Methodology for assessing modelled mixing ratios of chemical species in the UTLS

The Interpol-IAGOS software (Cohen et al., 2021) aims to facilitate the assessment of the model output with the IAGOS data
by deriving two respective products that are directly comparable. It consists of a projection of the scattered IAGOS data onto
the regular model grid, followed by a monthly average. For a given model, the subsequent gridded IAGOS product is then
denoted as IAGOS-DM-model, the -DM suffix referring to the distribution onto the model grid. For further simplicity, this
study refers to it as IAGOS–model. Concerning the model output, a daily mask is applied with respect to the IAGOS sampling
(Cohen et al., 2023), excluding the daily gridpoints with a sampling below a given threshold. This way, the subsequent monthly
products are representative of the same gridpoints and the same days. The seasonal and annual climatologies are then derived
from the monthly means with the same method and filtering as in Cohen et al. (2023).

The layers are defined as follow. The UT spreads from 400 hPa up to the isosurface of 2 PVU (potential vorticity units), but
excluding the top grid cell in order to avoid the strongest mixing zone, directly impacted by both layers (e.g. Thouret et al.,
2006; Cohen et al., 2018). The LS corresponds to all the sampled grid cells above the 3 PVU isosurface. In order to limit
the impact of errors in the modelled PV on the evaluation, we exclude the cases where the modelled PV value and the daily
average of the observed ozone mixing ratios are very likely to represent different layers. As in the previous studies (Cohen

et al., 2021, 2023), a daily grid cell in the UT (LS) is filtered out when the average ozone from IAGOS is more than 140 ppb (less than 60 ppb). Last, the non-separated UTLS represents all the measurement points above 400 hPa, and has been added into
this paper in order to include the output from the MOZART3 model without a PV field. In the tropics, the tropopause altitude does not allow the aircraft to sample stratospheric air masses. Consequently, only the upper troposphere is represented between 25° S and 25° N, and includes all the measurements above 300 hPa, though the uppermost part of the tropical troposphere remains higher than the cruise altitudes.

As in Cohen et al. (2023), the chosen metrics are the modified normalized mean bias (MNMB) with the fractional gross
error (FGE), and Pearson's correlation coefficient. Since the models have different PV fields and some of them have a different time periods, each model is compared to its own IAGOS–DM reference product. This study is divided into comparisons in the extratropics and in the tropics. The results in the extra-tropics are generally represented with several sets of metrics (quantiles, biases, and linear regression metrics). On the contrary, as the sampling in the tropics is more heterogeneous, three regions have been chosen as in Cohen et al. (2023) and are represented with mean zonal cross sections: West Atlantic–South America (called
South America hereafter), Africa and South Asia. Respectively, their zonal cross sections are averaged through the following longitude bands: 60° W–15° W, 5° W–30° E, and 60° E–90° E, as a compromise between an efficient sampling and a spatial uniformity for the observed species. For each of these regions, the year is divided into seasons that depend on the mean position of the intertropical convergence zone (ITCZ). In Cohen et al. (2023), the tropical season definitions were based on the months with the northernmost and southernmost position of the ITCZ (located with the observed horizontal winds and water vapour),
as an extension of Lannuque et al. (2021) that focused on Africa. They are respectively divided as: DJF–MAMJ–JA–SON, DJFM–AM–JJASO–N, and DJF–MAM–JJAS–ON.

## 3   Modelled reactive species compared to IAGOS observational data

This section is divided into two approaches. We first present introductory results showing global annual maps of model mean biases, then treat more precisely the northern extra-tropics in the UT and the LS separately and, to a lesser extent, in the
non-separated UTLS, and finally move into the (sub-)tropical UT characterization.

### 3.1   Horizontal distributions

The annual climatologies of the model biases in the UT and the LS are shown in Figs. 1–4 for the four models with an available PV field. A climatology is also shown for one of the IAGOS–DM products in order to provide an view of the expected features, but each bias remains relative to a different IAGOS–DM product. Since the IAGOS–DM climatologies are relatively similar



through the simulations with the same duration (not shown), we chose only one of the IAGOS–DM climatologies with the

longest time period (by default, IAGOS–EMAC). The mean biases in the non-separated UTLS for the five models are available

in Fig. A1. We can notice the sampling differences between the climatologies from MOZART3 (1997–2007), OsloCTM3

(2001–2017) and the other three models (1994–2017), notably in the tropical transects. Interannual variability is therefore

likely to cause differences in the observed climatologies especially for MOZART3, which period only includes six years of

frequent CO observations with IAGOS-CORE, a species with a great IAV due to biomass burning emissions (Voulgarakis et al.,

2015).

On yearly average, we first notice common features between the models. The biases in the non-separated UTLS (Fig. A1)

show an anticorrelation between ozone and CO in the northern extratropics, with a more pronounced stratospheric (tropo-

spheric) footprint in the high (low) latitudes, which is likely to reflect an overestimation (underestimation) of the tropopause

altitude. With the separation between the UT and the LS, the four models (now excepting MOZART3) clearly exhibit a geo-

graphical anticorrelation between ozone and CO biases in the LS (Figs. 1 and 2). The $O_3$/CO ratio shown in Fig. A2 summarizes

well this pattern by an MNMB decreasing with latitude. On the other hand, upper-tropospheric ozone is overestimated in the

mid-latitudes, whereas CO tends to be underestimated. These combined features suggest that the four models tend to overes-

timate the overall impact of stratosphere–troposphere exchanges across the extratropical tropopause. Most of the models also

overestimate ozone in the tropics, which is analysed in detail in Sect. 3.3. We can notice that the models showing a more

positive (negative) ozone (CO) bias in the low-latitude LS have the same tendency in the tropical UT, probably as an impact

of isentropic cross-tropopause exchanges. Figure 3 shows that contrary to the other species, each model $NO_y$ climatology is

simultaneously characterized by either low and high biases, but few grid cells with a weak bias. The map derived from the

observations is heterogeneous too, with upper-tropospheric minima over Northwest America, the North Atlantic corridor, and

near the Azores anticyclone (less than 400 ppt). A regional-scale maximum is visible over Northeast America (800–1400 ppt)

and over tropical Africa. The four models generally underestimate the magnitude of these geographical extrema. Last, in Fig.

4, the upper-tropospheric water vapour tends to be underestimated in the northern extratropics and, at least, in the northern

tropics.

## 3.2  Northern extra-tropics

In this subsection, we compare and characterize the observation and model seasonal cycles together, then show synthesizing

metrics to assess the model geographical distributions in the extra-tropics. Figures 5–8 provide an overview of the seasonal

climatologies in the UT and the LS, and Fig. 9 in the non-separated UTLS. Please note that the observed lower-stratospheric





**Figure 1.** Ozone mean horizontal distributions on annual averages from December 1994 until November 2017, for the IAGOS–EMAC product (left) and the biases for the masked output from EMAC, LMDZ–INCA, UKESM1.1, and OsloCTM3 (from left to right) normalized with respect to the mean values between each model output and their corresponding IAGOS–DM product, in the UT (bottom) and the LS (top).





**Figure 2.** Same as Fig. 1 for carbon monoxide, over the period 2002–2017.



**Figure 3.** Same as Fig. 1 for nitrogen reactive species, over the period 1997–2017 but with a more frequent sampling over the period 2001–2005.



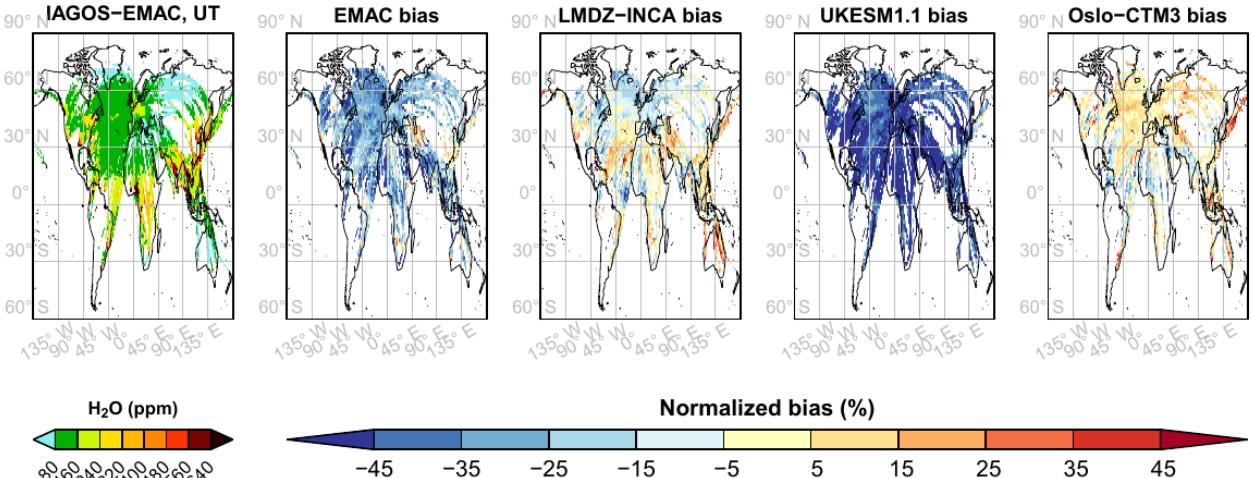

**Figure 4.** Same as Fig. 1 for water vapour in the upper troposphere.

water vapour in Fig. 5 is displayed as an indication for the seasonality, but due to the reasons explained in Sect. 2.3, part of its values are probably overestimated and it cannot serve for a bias quantification. The height of the boxplots illustrates the

geographical variability. The water vapour seasonality in the UT shown in Fig. 5 is well approximated in the simulations, with a wintertime minimum and a summertime maximum directly linked with convection and temperature that governs saturation vapour pressure. The lower stratosphere shows a similar pattern, though the contrast between the summertime water vapour maximum and the rest of the year is more pronounced. This feature is consistent with the increased impact from the troposphere during this season, and the extremely steep vertical gradient in water vapour. Based on CARIBIC measurements between 2005

and 2013, Zahn et al. (2014) found that the summertime maximum was primarily due to shallow cross-tropopause mixing in the extratropics. It takes place during all the year, thus the summertime $H_2O$ maximum in the UT increases the upward moisture flux. The other two pathways for moisture transport into the LS, i.e. localized deep convection events and, higher in the LS, the quasi-isentropic mixing with the tropical transition layer (TTL), mostly take place during summer (and early autumn for the latter) but were found to have a lesser contribution to the summertime moisture maximum.

The models have more difficulties in reproducing the CO seasonality. The observed springtime peak has been explained by an accumulation of CO in the lower troposphere during winter followed by an increase in the convective transport during spring (Cohen et al., 2018), then allowing the lower-tropospheric CO reservoir to impact the UT. This springtime maximum is not visible in the simulations, its magnitude is underestimated as well, and contrary to the observations, the springtime distribution is similar to winter, a feature that extends up to the LS. The comparison with the realistic water vapour cycles in



the UT tends to exclude convective transport from the causes of this discrepancy, except pyroconvection. It is possible that the
CO lifetime or the CO emissions are underestimated, which reduces the wintertime accumulation in the lower troposphere and
then the upward CO flux during spring. The lower values in the two tropospheric tracers ($H_2O$ and CO) in both UT and LS with
UKESM1.1 suggest an underestimation in the upward fluxes from the surface up to the UT, which favours an underestimation
in the LS too.

In the upper troposphere, the ozone maximum takes place in summer with a peak in photochemical activity, and the minimum
during winter. In the lowermost stratosphere, the ozone maximum takes place during spring with the effects of the descending
branch of the Brewer–Dobson circulation, transporting ozone rich air masses down from the deeper extratropical stratosphere,
whereas the minimum takes place during fall. The models reproduce well these features, but also the dichotomy in the UT
between high-ozone seasons (spring and summer) and low-ozone seasons (winter and fall). Though most models are positively

biased, the magnitude of the seasonal cycle is similar with the observations. In the LS, the ozone distribution is harder to repro-
duce during summer and fall, with a geographical variability only spreading over the lower half of the observed distribution.
It is characterized by a tendency to overestimate the low-latitude ozone minima and underestimate the high-latitude ozone
maxima (e.g. Fig. 1). Similarly with ozone, both observed and modelled $NO_y$ mixing ratios show a springtime maximum in
the LS and a summertime maximum in the UT. In the UT, the summertime maximum is linked to the photochemical activity, to

the enhanced lightning frequency, to more intense boreal forest fires and to an enhanced convection that uplifts diverse ozone
precursors from the lower troposphere. It is consistent with the detailed individual $NO_y$ species in Figs. B1–B3, where each of
the models generally shows a summertime maximum in $NO_x$, PAN and especially $HNO_3$. The latter is notably affected by the
conversion from $NO_x$ with the photochemical activity (Stratmann et al., 2016). In the LS, the impact of the Brewer–Dobson
circulation coupled with the $HNO_3$ production from the nitrous oxide decomposition in the stratosphere is reproduced, as

shown in Fig. B3. The only exception is the UKESM1.1 model that rather shows an upper-tropospheric seasonality in the
LS, though the influence of the Brewer–Dobson circulation remains visible through more elevated springtime mixing ratios
compared to the UT seasonal cycle. On the contrary, the OsloCTM3 model shows higher $NO_y$ values in the LS. As the ozone
amounts are within the same range as the other models, it excludes the stratospheric circulation from the possible causes of
the $NO_y$ discrepancies. For UKESM1.1 (OsloCTM3), the lower (higher) $HNO_3$ values in the LS (Fig. B3) might thus be

due to an underestimation (overestimation) of the $N_2O$ flux into the stratosphere, to an overestimated (underestimated) $N_2O$
lifetime in the stratosphere, or to an underestimated (overestimated) $HNO_3$ lifetime against stratospheric aerosol uptake. The
latter is a possible contributor with OsloCTM3, as its mass density of particular sulfate and nitrate is lower by 10 % than in





**Figure 5.** Boxplots synthesizing the mean geographical distribution in extratropical water vapour in the LS (top) and the UT (bottom), for the IAGOS–EMAC product and for the model products (from left to right). Each color corresponds to a season, the black boxes representing the annual means. For a given boxplot, the white line represents the median, the box corresponds to the interquartile interval, and the whiskers represent the values between the 5 and 95 percentiles. Please note that due to its uncertainty, the observed $H_2O$ in the LS shown here cannot be used for an accurate quantification, hence the dashed boxplots.





**Figure 6.** Same as 5 for CO.



the LMDZ-INCA simulation. Considering both ozone and $NO_y$ in the LS, the similarities between observations and models, notably during spring, are encouraging regarding the stratospheric chemistry and diabatic transport for all the models.

As shown in Figs. B1–B2 and Fig. B4, the $NO_y$ partitioning changes substantially between the models. In the UT, a higher proportion of $NO_y$ is represented by $HNO_3$ with OsloCTM3. As it is affected by wet scavenging, it can explain the lower $NO_y$ mixing ratios with this model, combined with very low PAN quantities. On the contrary, the higher levels of $NO_y$ in the UT with the EMAC model can be linked to the high proportion of PAN which is not soluble and has a several-month chemical lifetime (e.g. Fadnavis et al., 2015). The higher amounts of PAN themselves might be linked to the EMAC colder temperatures

$(\sim -4K)$, as it increases its lifetime against thermolysis. In the UT, the higher (lower) $NO_x$ mixing ratios from the EMAC (LMDZ-INCA) model can also explain the higher (lower) ozone mixing ratios. The intermodel variability regarding PAN in the UT might also have consequences on the air quality evaluation in the subsidence regions, as the PAN lifetime against thermolysis decreases drastically during the descending motion, from months down to minutes when the temperature reaches 20 °C. Regarding this variability, the surface ozone production due to the PAN subsidence is thus likely to vary substantially

across the models (at least in the remote areas with a $NO_x$-limited regime), with a maximum for EMAC and a minimum for OsloCTM3. In the LS, the PAN intermodel variability is the most noticeable of all the $NO_y$ species, with a factor reaching 12 between the median mixing ratio from EMAC and OsloCTM3, and an important difference in every couple of models. The low (high) amounts with OsloCTM3 (EMAC) are, at least partially, related to the low (high) amounts in the UT as well.

     The lower-stratospheric features described above are generally visible in the UTLS as well, as illustrated in Fig. 9, for the

species with a strong positive vertical gradient. Notably, the springtime maximum is well represented by every model for ozone and almost all the models for $NO_y$, including MOZART3, which confirms that all the models catch the seasonality of the Brewer–Dobson circulation. The water vapour and temperature maximum in summer is also visible in the simulations. For the five models, the $NO_y$ seasonal cycle is characterized by a springtime maximum in $HNO_3$ due to the Brewer–Dobson circulation, and by a summertime maximum in both $NO_x$ and PAN due to convection, photochemistry and lightning emissions.

In the previous figures (Figs. 7–9), we can notice that when the annual means are noticeably biased, the sign of this bias is generally representative of all the seasons, though its magnitude is not. With this perspective, the annual means shown below still provide relevant information. Figure 10 synthesizes some model skills in terms of annual averages in the extra-tropics, and the intermodel ranges are indicated more precisely in Table 3. The correlation coefficient is generally greater in the non-separated UTLS than in the UT or the LS, for the variables with an important vertical gradient (ozone, CO, $O_3$/CO, $NO_y$).

As mentioned in Cohen et al. (2023), this difference suggests at least that the most important changes in the mean tropopause height are well represented, notably the meridian gradient as shown in Fig. C1. In the separated layers, the ozone mixing ratio





**Figure 7.** Same as 5 for ozone.



**Figure 8.** Same as 5 for $NO_y$.




**Figure 9.** Boxplots synthesizing the mean geographical distribution in extratropical variables (from top to bottom: ozone, CO, $NO_y$, water vapour and temperature) in the non-separated UTLS, for the IAGOS–EMAC product and for the five model products (from left to right). It is worth reminding that the IAGOS–MOZART3 product has substantially different distributions, so the MOZART3 climatologies cannot be compared directly to the IAGOS–EMAC product. Please note that due to its uncertainty, the observed $H_2O$ in the LS shown here cannot be used for an accurate quantification, hence the dashed boxplots.





is generally more difficult to model in the UT than in the LS, in terms of geographical distribution ($r_{UT}(O_3)$ = 0.45–0.74, compared to $r_{LS}(O_3)$ = 0.75–90) as in terms of mean biases, the latter being essentially positive for most models in the UT (MNMB = 0.005–0.36, FGE $\sim$ MNMB for three models), and weak in the LS (MNMB = -0.08–0.006, with a maximum FGE

at 0.16, particularly low). On the contrary, the CO correlation coefficient is higher in the UT ($r_{UT}(CO)$ = 0.63–0.78) than in the LS ($r_{LS}(CO)$ = 0.11–0.70), probably reflecting a difficulty in mapping the effects of the cross-tropopause exchanges. In the UT, the ozone and CO biases tend to be respectively positive and negative. This antagonism can be linked with overestimated cross-tropopause exchanges and/or an overestimated photochemical activity, thus with more ozone production causing more CO destruction. In the UT as well, both surface tracers (CO and $H_2O$) show good correlations ($r_{UT}(H_2O) \sim 0.95$ and, for most

models, $r_{UT}(CO) \sim 0.8$). The skill difference between the two species can be explained either by uncertainties in CO emissions in each region, or an underestimation of the detrainment altitude from pyroconvection. Interestingly, the bias magnitudes for both species are higher for EMAC and UKESM1.1. Concerning EMAC, the systematically negative temperature biases (-3.7 K on average) could be another factor controlling the lower water vapour amounts in the UT via saturation, but it would not be consistent with the combination of lower temperatures (-4.0 K on average) with more water vapour in the LS, compared

to the other models. A comparable cold bias with EMAC has been diagnosed in Righi et al. (2015) at 200 hPa with a similar simulation set-up, who also identified a wet bias compared to the observations from the Halogen Occultation Experiment (HALOE: Grooß and Russell III, 2005), at 200 hPa in the extra-tropics. They concluded that an overestimation of lower-stratospheric water vapour would cause an excessive radiative cooling and thus a cold bias, a relation that had already been shown in previous studies. This moisture overestimation in the LS is confirmed in Fig. 5, showing higher $H_2O$ amounts in

EMAC compared to the observations, given that the latter are probably overestimated. Last, $NO_y$ shows the largest variability in the MNMB in the UT, and the lowest correlation coefficient among the four chemical species. It can be due to the important intermodel variability in the spatial distribution of lightning emissions (e.g. Hakim et al., 2019) or to the washout of $HNO_3$, which depends on the cloudiness representation and on the $NO_y$ partitioning.

Figures 11–13 provide further information on the annual geographical distribution mentioned above for each species, layer

and model, in the northern extra-tropics. The particularly high correlation for water vapour (r = 0.95) shown in Fig. 11 is characterized by a well-reproduced meridional structure, notably with the strong variability in the lowest latitudes (orange and red dots) due to dry subsiding and moist convective regions. It is notably characterized by a linear regression slope close to 1 for EMAC, LMDZ-INCA and OsloCTM3. All of these features are representative of all the seasons, with the highest correlation and linear regression slope during summer, when the tropospheric humidity reaches its maximum. These features



**Figure 10.** Modified Taylor diagrams synthesizing the assessment the yearly climatologies beyond $25°$ N derived from the five model output against their respective IAGOS–DM product, for $O_3$, CO, $O_3$/CO ratio, $NO_y$, and upper-tropospheric $H_2O$. Each model is represented by a color, and each layer by a point shape. The radial axis corresponds to the modified normalized mean bias (MNMB) for the chemical compounds, and the orthoradial axis refers to the r correlation coefficient. The error bars are the quartiles 1 and 3 of the normalized biases shown in Figs. 1–4.



**Table 3.** Annual metrics synthesizing the assessment of the $O_3$, CO, $O_3$/CO ratio, $NO_y$, $H_2O$, and temperature climatologies from the model simulations against their respective IAGOS–DM product in several layers, as shown in Fig. 10 and Figs 11–13. From left to right: the Pearson correlation coefficient (r), the modified normalized mean bias (MNMB), the fractional gross error (FGE) and the sample size ($N_{cells}$). For the temperature, the absolute bias and its associated error are equivalent to the MNMB and the FGE without the normalizing factors. Each metric is represented by an intermodel range.

| Species | Layer | r | MNMB | FGE | $N_{cells}$ |
|---------|-------|---|------|-----|-------------|
| $O_3$ | UTLS | 0.83–0.96 | [-0.04, 0.14] | 0.10–0.25 | 4,080–5,361 |
|  | LS | 0.75–0.90 | [-0.08, 0.006] | 0.11–0.16 | 4,368–4,604 |
|  | UT | 0.45–0.74 | [0.005, 0.36] | 0.07–0.36 | 3,144–3,577 |
| CO | UTLS | 0.47–0.91 | [-0.33, 0.12] | 0.07–0.33 | 3,982–5,433 |
|  | LS | 0.11–0.70 | [-0.36, 0.23] | 0.09–0.36 | 4,458–4,702 |
|  | UT | 0.63–0.78 | [-0.31, -0.08] | 0.09–0.31 | 3,145–3,636 |
| $O_3$/CO | UTLS | 0.73–0.92 | [-0.33, 0.36] | 0.24–0.39 | 3,918–5,164 |
|  | LS | 0.69–0.83 | [-0.41, 0.32] | 0.26–0.42 | 4,135–4,470 |
|  | UT | 0.39–0.54 | [ 0.04, 0.56] | 0.12–0.56 | 2,778–3,260 |
| $NO_y$ | UTLS | 0.63–0.79 | [-0.44, 0.25] | 0.19–0.475 | 3,645–3.894 |
|  | LS | 0.49–0.66 | [-0.58, 0.40] | 0.17–0.58 | 3,077–3,274 |
|  | UT | 0.50–0.62 | [-0.14, 0.28] | 0.32–0.38 | 1,831–2,187 |
| $H_2O$ | UT | 0.94–0.96 | [-0.47, 0.025] | 0.11–0.47 | 3,289–3,642 |
|  |  |  | Abs. bias (K) | Abs. error (K) |  |
| T | UTLS | 0.95–0.97 | [-3.8,-0.4] | 0.7–3.9 | 4,172–5,759 |
|  | LS | 0.79–0.86 | [-4.0,-0.4] | 0.7–4.0 | 4,952–5,132 |
|  | UT | 0.98–0.99 | [-3.7,-0.5] | 0.8–3.7 | 3,646–4,002 |

are encouraging concerning the modelled impact of meteorological systems on the extratropical UT in terms of geographical variability, despite the negative mean biases present from most models.

In the lowermost stratosphere, Fig. 12 shows that the ozone geographical variability is relatively well reproduced by the models with a distinct northward gradient. This gradient tends to be underestimated because of a positive bias in the lowest values (in the subtropics) for most models, and a negative bias in the highest values in the subpolar regions. The situation is

similar for $NO_y$ (Fig. 13) but with a substantially lower correlation coefficient. Contrary to ozone, this is characterized by poor correlations inside each zonal band, and suggests that the $NO_y$ correlation in the LS is mostly due to the northward gradient and that the smaller scales are hardly captured by the models for this variable.





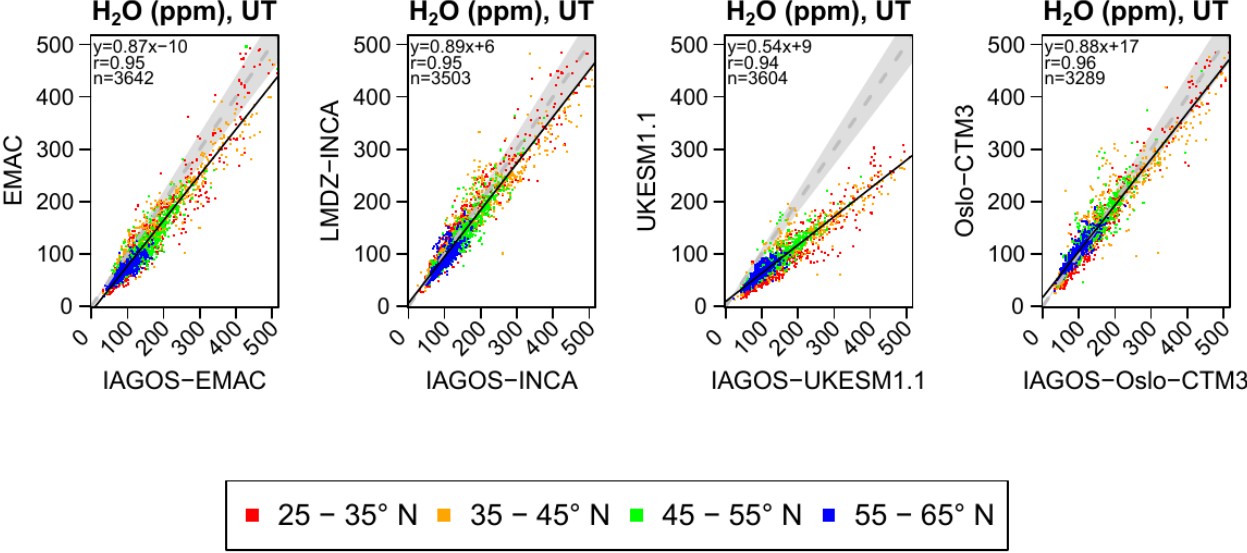

**Figure 11.** Scatterplots comparing the geographical distributions in water vapour in the extratropical UT between the models output (Y axis) and their respective IAGOS products (X axis), in terms of annual means. The colors display a latitude band, from subtropical (red) so subpolar latitudes (blue).





**Figure 12.** Same as Fig. 11 with ozone, in the LS (top panels) and in the UT (bottom panels).





**Figure 13.** Same as Fig. 12 for $NO_y$.

It has to be noticed that these diagrams average values through the whole sampled extra-tropics in the northern hemisphere. An assessment based on regional specific characteristics could provide more information on specific processes, as tropopause

folds during the Middle East summer or the isentropic transport from the tropical troposphere into the extratropical lowermost stratosphere (Cohen et al., 2018). Another limitation of this approach is that the mean altitude of the measurements changes with the latitude: as the tropopause altitude decreases with the latitude, the subtropics are more sampled in the UT than in the LS, and reversely, the high latitudes are more sampled in the LS than the UT. The scores shown for the different layers are thus not completely representative of the same geographical area.



The comparison with previous model assessment studies provides complementary information. First, most of the CCMVal-2 models assessed in Hegglin et al. (2010) underestimated the vertical stability in the northern mid-latitudes (40–60° N), especially for the semi-Lagrangian models and the models with the lowest vertical resolution. As a consequence, they generally underestimated ozone and $HNO_3$, and overestimated water vapour at 200 hPa, which is included in the lowermost stratosphere at these latitudes. Though all the CCMVal-2 models did not have a specific tropospheric chemical scheme, and though only the

EMAC model is involved in both studies, our results tend to confirm the ability of the models to reproduce the seasonality of the Brewer–Dobson circulation through the ozone and $HNO_3$ tracers, and the overestimation of the cross-tropopause mixing, notably the effect of the tropospheric influence on the lower-stratospheric ozone that maximizes during summer and fall. On this last point, the effect of the vertical resolution is visible on lower-stratospheric ozone with the less resolved model (LMDZ-INCA) showing the lowest ozone vertical gradient. Still, it does not seem to be the most controlling factor for water vapour, as

the EMAC model is one of the most resolved models but has the weakest water vapour vertical gradient.

In addition to the models assessment, the model intercomparison of background CO, ozone and $NO_x$ in the UT and the LS (Figs. 6, 7, and B1 respectively) can provide further understanding of each model's ozone sensitivity to aircraft $NO_x$ emissions, as the critical NO mixing ratio separating net production and net destruction of ozone depends on these three parameters (Grooß et al., 1998). It remains uncertain as it ignores the behaviour of lots of non-measured VOCs and methane, but it still provides

a comparison of several factors controlling the sensitivity of the net ozone production to the $NO_x$ emissions. In the UT, we can expect the most different ozone responses between the EMAC and LMDZ-INCA models, as EMAC shows higher $NO_x$ and ozone values and lower CO values, contrary to LMDZ-INCA. In the LS, it can also be expected that the LMDZ-INCA model maximizes the ozone response since $NO_x$ (CO) are relatively low (high), and this difference can be enhanced during summer and fall with relatively lower ozone values. The two models showing the highest $NO_x$ values (EMAC and especially

OsloCTM3, with more than twice the median compared to LMDZ-INCA and UKESM1.1) can be expected to have a lower ozone response.

### 3.3   Tropics

The zonal cross sections shown in Figs. 14 and 15 compare the reference runs with the observations in three tropical regions: South America–Atlantic (called South America hereafter), Africa and South Asia. First, we present a brief summary of some

observed patterns that have been investigated in Cohen et al. (2023), notably based on Livesey et al. (2013), Lannuque et al. (2021) and Gottschaldt et al. (2018) for the three regions, respectively from west to east. In a second step, we present an overview of the model skills. It is worth noticing the changes in the mean pressure (represented at the top of the graphics





for the longest climatologies), sometimes correlated with changes in the observed variable. Another relevant point is that the differences in the sampling period cause differences in the IAGOS–OsloCTM3 transects (2001–2017 instead of 1995–2017) in

ozone (July–August in South America) and water vapour (June–October above Africa). We thus chose to represent the IAGOS–OsloCTM3 profiles as well. It is also the case with the IAGOS–MOZART3 transects (1997–2007) regarding CO above Africa in December–March, but the subsequent lessened amount of data added to the impossibility to filter out the stratospheric air masses with the same criterion as the other models (PV) results in shorter meridian profiles, or even not shown.

     Most of the observed features have been investigated in Cohen et al. (2023), as the impacts of wet and dry seasons linked

to the shifts in the inter-tropical convergence zone (ITCZ). In the two western regions (South America and Africa) where the zonal cross sections cover most of the tropical latitudes, the wet season is characterized by a collocated maximum in water vapour and a minimum in ozone, both linked with intense convection of humid surface air. The latter is rather rich in fresh pollutants and gets enriched in $NO_x$ emitted by lightning during their convective uplift. In the upper branch of both Hadley cells, ozone is produced by photochemistry during its poleward transport.

Above Africa, the seasons with the northernmost and the southernmost ITCZ (resp. June–October and December–March) are particularly visible in the IAGOS observations. During the meridional transport in the upper branch of the strongest Hadley cell, CO accumulates near the wind shear areas (Sauvage et al., 2007; Lannuque et al., 2021) as well as $NO_y$, reaching a maximum at approximately $10°$ away from the ITCZ. Using a method based on the FLEXPART Lagrangian dispersion model (SOFT-IO: Sauvage et al., 2017), Lannuque et al. (2021) found that these CO peaks were originated from intense biomass

burning in the dry season. A sensitivity test regarding biomass burning with the LMDZ-INCA model (Cohen et al., 2023) found similar conclusions. In the monsoon season (June–October), both studies agree on a major biomass burning contribution to the southward shift of the CO peak. Last, the Asian Summer Monsoon (right panels) is characterized by the warmest and most humid air masses, as expected from the most intense convective system. During this season, the subtropical jet stream and its subsequent stratospheric intrusions are confined on the northern side of the Himalayas (Cristofanelli et al., 2010), thus

ensuring a weak stratospheric influence in this season (Gottschaldt et al., 2018).

     The analysis of the modelled local and seasonal features provides interesting information about the representation of the convective systems and their outflows. All the models capture the location of the peaks in water vapour in the African upper troposphere whatever the season, the maximized water vapour amounts (as well as temperature, not shown) above the Asian summer monsoon (Fig. 15, right column), and the strongest CO peaks above Africa. More precisely above Africa, the

December–March (June–October) season, the models capture relatively well the southward (northward) shift of the ITCZ, characterized by an ozone minimum collocated with the water vapour maximum. This agreement among the models and with







**Figure 14.** Zonal cross sections between 25° S and 30° N from December until February or March. Each row represents a measured variable, and each column represents the corresponding region: from left to right, South America–Atlantic ocean, Africa, and South Asia. The uncertainties shown here correspond to the spatial variability, defined as the interval between the quartiles 1 and 3. The solid black, dark-green and dark violet lines correspond respectively to IAGOS–EMAC, IAGOS–Oslo and IAGOS–MOZART. For more visibility, the observational variability is shown only for the IAGOS–EMAC profiles. The blue, red, orange, green and violet lines correspond respectively to the reference simulation from EMAC, LMDZ-INCA, UKESM1.1, OsloCTM3, and MOZART3. The dashed line at the top of each graphic shows the mean pressure derived from IAGOS–EMAC. Its values are reported on the right axis.





**Figure 15.** Same as Fig. 14 for July–August, June–October and June–September, from left to right. Please note the different scale for water vapour in the right column.



the observations highlights a realistic representation of the most convective systems. It is probably favored by the nudging, by the use of a common surface temperature field based on observations, and by a common (or similar) inventory for biomass burning emissions.

Regarding the effects of convection, the variability between the models can be found in the water vapour and CO peaks intensity, and the CO peak location. In most cases, water vapour shows a small bias with LMDZ-INCA and OsloCTM3, and a dry (moist) bias with EMAC and UKESM1.1 (MOZART3). The EMAC dry bias is possibly explained by a cold bias in the UT ($\sim$ -5 K, not shown) lowering the saturation vapour pressure and/or the detrainment altitudes. The other models show particularly well reproduced temperatures, and all the modelled temperature profiles are well correlated with the observations
(not shown).

The CO maximum location and width depends on the model, notably above Africa: it is rather collocated with the ITCZ for the EMAC and UKESM1.1 models, and shifted 5–10° equatorward for the other models, in agreement with the observations. Concerning June–October, with the same observation data set, Lannuque et al. (2021) show a peak in anthropogenic contribution to upper-tropospheric CO collocated with the ITCZ (as with EMAC and, to a lesser extent, UKESM1.1 and
OsloCTM3) and a peak in biomass burning contribution shifted 10° southward, as with the LMDZ-INCA and MOZART3 models. OsloCTM3 seems to show a compromise between the two categories, with a flatter and wider maximum including both the ITCZ position and the observed CO peak. A sensitivity test (Cohen et al., 2023) using the LMDZ-INCA model that reproduces well the CO peak during December–March and June–October (with its location as well as its magnitude near 140 ppb) concluded on the biomass burning being the main factor of the peak intensity with a contribution reaching 30 and 45 ppb
during DJFM and JJASO respectively, but also in the southward CO shift during June–October. The negative CO bias combined with the absence of the southward shift in the EMAC and UKESM1.1 simulations is thus likely to reflect an underestimation of the impact from biomass burning in the tropical UT. As the dry bias in the water vapour peaks in these two models suggests a less intense convection, it implies a weaker Hadley circulation. Consequently, the lower-tropospheric entrainment into convective motions in the ITCZ has a reduced geographical extent and thus includes less air from the dry region. It could explain
the lack of CO accumulation in the higher tropical latitudes with these two models. Inversely, the LMDZ-INCA model and, to a lesser extent OsloCTM3, show another peak in CO in July–August above South America at relatively similar latitudes as for the peak in Africa, absent from the observation profiles. As it is mainly due to biomass burning for LMDZ-INCA (Cohen et al., 2023), and as LMDZ-INCA and OsloCTM3 show similar behaviours with CO, it suggests that both models overestimate the effects from the intercontinental connection with Africa during this season (in duration and/or in intensity) or from local
biomass burning emissions.

Contrary to CO, the peaks in $NO_y$ observed in December–February (December–March) show an important negative bias in LMDZ-INCA and OsloCTM3, whereas the negative bias is lessened with EMAC and UKESM1.1, especially above South America. Concerning the Asian summer monsoon, all the represented models overestimate the ozone and $NO_y$ mixing ratios, possibly reflecting an overestimation of the lightning flash rate (as it is known for LMDZ-INCA), an underestimation of the

$HNO_3$ uptake, and/or an overestimation of the entrained surface pollutants.

Most of the models tend to overestimate ozone, except the LMDZ-INCA model that rather shows negative biases. The ozone overestimation in the tropical UT from UKESM1.1 is consistent with a recent comparison based on ozone partial columns between UM–UKCA and the OMI–MLS satellite observations (Russo et al., 2023) during the 2005–2018 period and in the 450–170 hPa pressure range. The overestimation is representative of the whole tropospheric ozone column in the tropics, and

the main factor in the UT is probably an overestimation of the lightning $NO_x$ emissions. The LMDZ-INCA model shows particularly low $NO_x$ levels with lowest (highest) mean values near 18 ppt (195 ppt) whereas the other models have $NO_x$ minimum (maximum) levels at 52–91 ppt (278–430 ppt). This first order statement can be sufficient to explain most of the LMDZ-INCA lower ozone values, but also most of the higher CO values with longer photochemical lifetimes for CO. Each of these two factors favour the ozone production efficiency from lightning and aviation. A similar diagnostic applies to $HNO_3$.

Though the stronger convection with LMDZ-INCA compared to EMAC theoretically produces more $NO_x$ due to a higher lightning activity, it is thus possible that the LMDZ-INCA model overestimates the $NO_y$ removal by $HNO_3$ wet scavenging, both with a more efficient conversion of $NO_x$ into $HNO_3$, and with further precipitation due to a stronger convection.

## 4    Conclusion

The present study consists of a descriptive evaluation of 5 global chemistry-climate and chemistry-transport models (CCMs

and CTMs) against the airborne IAGOS observations. The assessment is based on ozone, carbon monoxide (CO), water vapour ($H_2O$), reactive nitrogen ($NO_y$) and, to a lesser extent, temperature. It relies on measurements during the cruise phases, i.e. in the extratropical upper troposphere–lower stratosphere (UTLS) and in the tropical upper troposphere.

A direct comparison between the model outputs and the IAGOS data set is made possible with the use of the Interpol-IAGOS software, which projects the IAGOS data onto the model grid with a daily resolution (Cohen et al., 2023). Meanwhile, a daily

mask is applied to the model output with respect to the IAGOS sampling. For each grid cell, a weighted monthly average is then derived from both gridded observations and model output. For a given model, the subsequent IAGOS and model products are called IAGOS-DM-model and model-M respectively (the –DM and –M suffixes referring to the distribution onto the model grid, and to the IAGOS mask respectively), and are directly comparable between each other. In the extra-tropics, the model





potential vorticity (PV) is used to treat separately the upper troposphere (UT) and the lower stratosphere (LS). The assessment
is based on the climatologies derived from these products, between 1995 until 2017 for most models. A synthesis of the model
skills in reproducing the main observed atmospheric features is proposed in Table 4.

In the northern mid-latitudes, the results suggest that most models tend to overestimate the cross-tropopause mixing. The
stratospheric tracers ($O_3$ and, to a lesser extent, $NO_y$) tend to be overestimated in the UT and underestimated in the LS.
Concerning the tropospheric tracers (CO and $H_2O$), all the models systematically underestimate CO in the UT whereas only
two of them systematically underestimate water vapour in this layer. It would be consistent with an underestimation of the
CO emissions from the CEDS inventory, and/or with an overestimation of the CO photochemical loss. The geographical
distributions are particularly well correlated with observations for ozone in the LS, and water vapour in the UT. Respectively,
the former and the latter suggest a realistic distribution of the impacts from the stratospheric circulation, and of the synoptic-
scale processes in the troposphere. The impact from biomass burning and lightning is harder to reproduce, notably because of
the difficulty of parameterizing pyroconvection, lightning and the washout of soluble species.

The seasonality is generally consistent between models and observations, both in the UT, the LS, and the non-separated
UTLS. Discrepancies are visible with CO in the UT, and ozone in the LS. The former is characterized by a seasonal maximum
gathering winter and spring contrary to the observed springtime maximum, and with an important negative bias in spring,
which may suggest an underestimation of the CO emissions in winter and spring, as it concerns all the models. Ozone shows
a stronger summertime decrease in the models than in the observations, probably caused by an overestimated influence from
the troposphere, particularly during summer and fall. For each season, the models tend to underestimate the geographical
variability in every measured species. One possible consequence is an excess of horizontal homogeneity in the ozone response
to aircraft $NO_x$ emissions, but it is hard to conclude as the background $NO_x$ cannot be compared with the observations in the
same way as the other species.

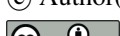



**Table 4.** Synthesis of the model abilities to reproduce the main features from their respective IAGOS–DM products, regardless to their mean biases.

| Layer | Species | Main features from IAGOS–DM | EMAC | LMDZ–INCA | Reproduced by: UKESM1.1 | OsloCTM3 | MOZART3 |
|---|---|---|---|---|---|---|---|
| LS* | O$_3$ | Springtime maximum | Yes | Yes | Yes | Yes | Yes |
| | | Northward gradient | Yes | Yes | Yes | Yes | Yes |
| | NO$_y$ | Springtime maximum | Yes | Yes | No | Yes | Yes |
| | | Northward gradient | Yes | Yes | Yes | Yes | Yes |
| | H$_2$O | Summertime maximum | Yes | Yes | Yes | Yes | Yes |
| | CO | Summertime maximum | No | Yes | Yes | Yes | - |
| | | Southward gradient | No | Yes | Yes | Yes | No |
| UT (extra-tropics) | O$_3$ | Summertime maximum | Yes | Yes | Yes | Yes | - |
| | NO$_y$ | Summertime maximum | Yes | Yes | Yes | Yes | - |
| | H$_2$O | Summertime maximum | Yes | Yes | Yes | Yes | - |
| | | Southward gradient | Yes | Yes | Yes | Yes | - |
| | | High variability at low lat. | Yes | Yes | Yes | Yes | - |
| | CO | Springtime maximum | No | No | No | No | - |
| (tropics) | O$_3$ | ITCZ minimum | Yes | Yes | Yes | Yes | Yes |
| | NO$_y$ | Boreal winter: high south-north difference | Yes | No | Yes | Yes (Africa) | - |
| | H$_2$O | ITCZ maximum | Yes | Yes | Yes | Yes | Yes |
| | | More H$_2$O in the ASM** | Yes | Yes | Yes | Yes | Yes |
| | CO | Africa: max. shifted from the ITCZ | No | Yes | No | Yes | Yes |

**Table 4.** *Or UTLS for MOZART3, if the feature is also visible in the UTLS with IAGOS–DM.
**ASM: Asian Summer Monsoon.

The intermodel variability is particularly noticeable with the NO$_y$ individual species, in the UT as in the LS. The median NO$_x$ level varies by a factor up to 3 in the UT, and up to 7 in the LS. It reflects both different chemical and physical behaviours, as the NO$_x$ conversion into HNO$_3$ or PAN, and the HNO$_3$ wet scavenging that removes gaseous NO$_y$ from this atmospheric region, or also the aerosol uptake of HNO$_3$. It has implications on the model sensitivity to the NO$_x$ injection in the UTLS from subsonic aviation as it changes the NO$_x$ regime, and on the evaluation of air quality from the models in the subsidence regions as PAN varies substantially across the models and in rapidly converted into NO$_x$ at the typical surface temperatures.

The addition of the NO$_x$ measurements from CARIBIC will allow an evaluation of NO$_x$ biases, at least in the most sampled regions. On a longer term, the IAGOS-CORE measurements of NO$_x$ will open the opportunity of calculating NO$_x$ climatologies as well, with a higher level of sampling. The particulate matter measurements will also provide another variable for the assessment, and further explanation for the chemical processes related to HNO$_3$. Concerning the models, a more accurate interpretation of the inter-model variability could be provided with additional variables such as horizontal wind velocities, potential temperature, and inert tropospheric and stratospheric tracers, in order to isolate further the role of dynamics or chemistry in



the modeled mixing ratios. In the extra-tropics, the choice of more accurate dynamical coordinates as the equivalent latitudes (involving both potential temperature and PV) or the jet-related tropopause (Millán et al., 2024) will probably improve the model assessment. Last, the model abilities to simulate the long-term trends have to be evaluated as well, as a complementary

part of the current analysis.



## Appendix A: Geographical distributions



**Figure A1.** Biases in the non-separated UTLS from the five models compared to their respective IAGOS–DM products. From top to bottom, the variables are ozone, CO, NO$_y$ and temperature. The temperature biases are not normalized.



**Figure A2.** Same as Fig. 1 for the $O_3$/CO ratio, over the period 2002–2017.



## Appendix B: Individual $NO_y$ species



**Figure B1.** Boxplots synthesizing the contribution of $NO_x$ to the $NO_y$ levels shown in Fig. 8 in the LS (top) and the UT (bottom), for the four model products.





**Figure B2.** Same as Fig. B1 for peroxyacetyl nitrate (PAN).







**Figure B3.** Same as Fig. B1 for nitric acid ($HNO_3$).





**Figure B4.** Boxplots synthesizing the contribution of $NO_x$, $HNO_3$ and PAN to the $NO_y$ levels shown in Fig. 9 in the non-separated UTLS, for the five model products. The upper panel is the same as in Fig. 9.



**Appendix C: Scatterplots in the non-separated UTLS**

**Figure C1.** Scatterplots comparing the vertical averages for the models and the IAGOS–DM products in the non-separated UTLS and on an annual average, for $O_3$, CO, $NO_y$, and temperature (from top to bottom).



# Appendix D: Seasonal assessment of modelled reactive species in the northern extra-tropics

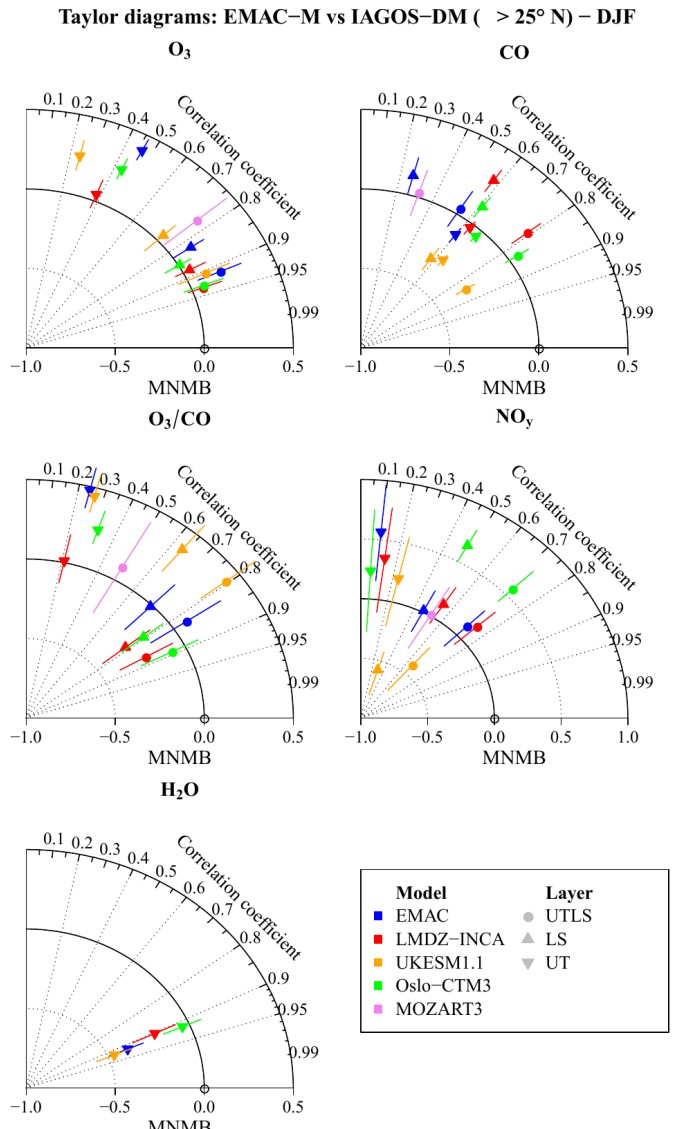

**Figure D1.** Same as Fig. 10 for boreal winter.



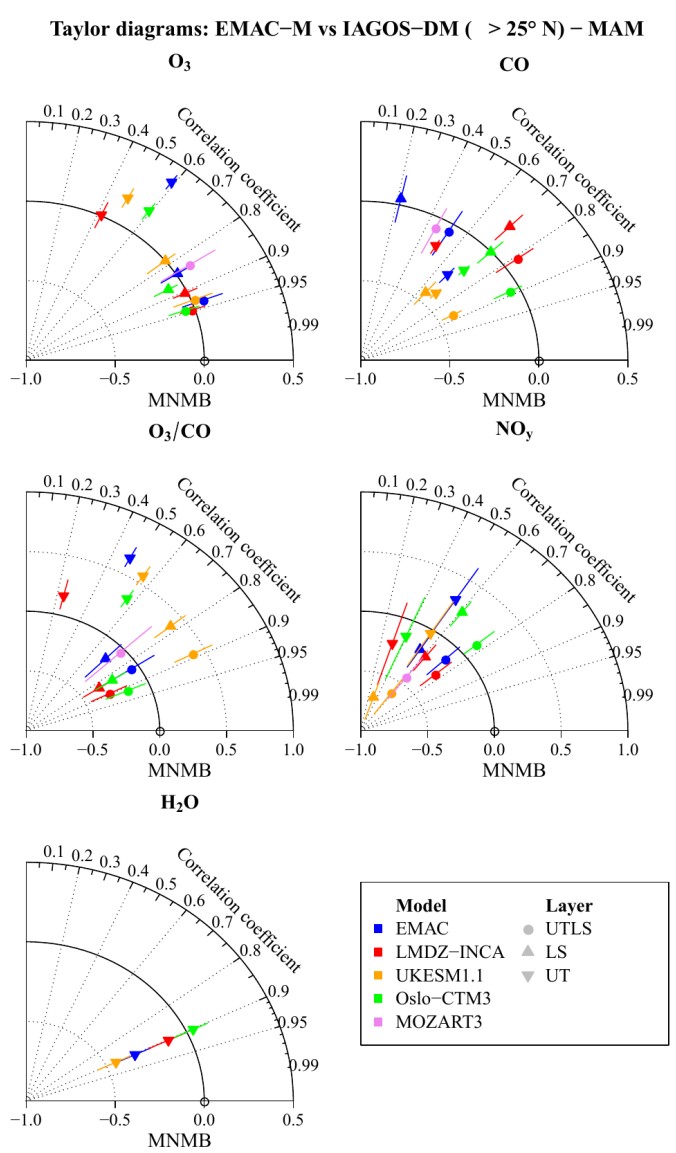

**Figure D2.** Same as Fig. 10 for boreal spring.



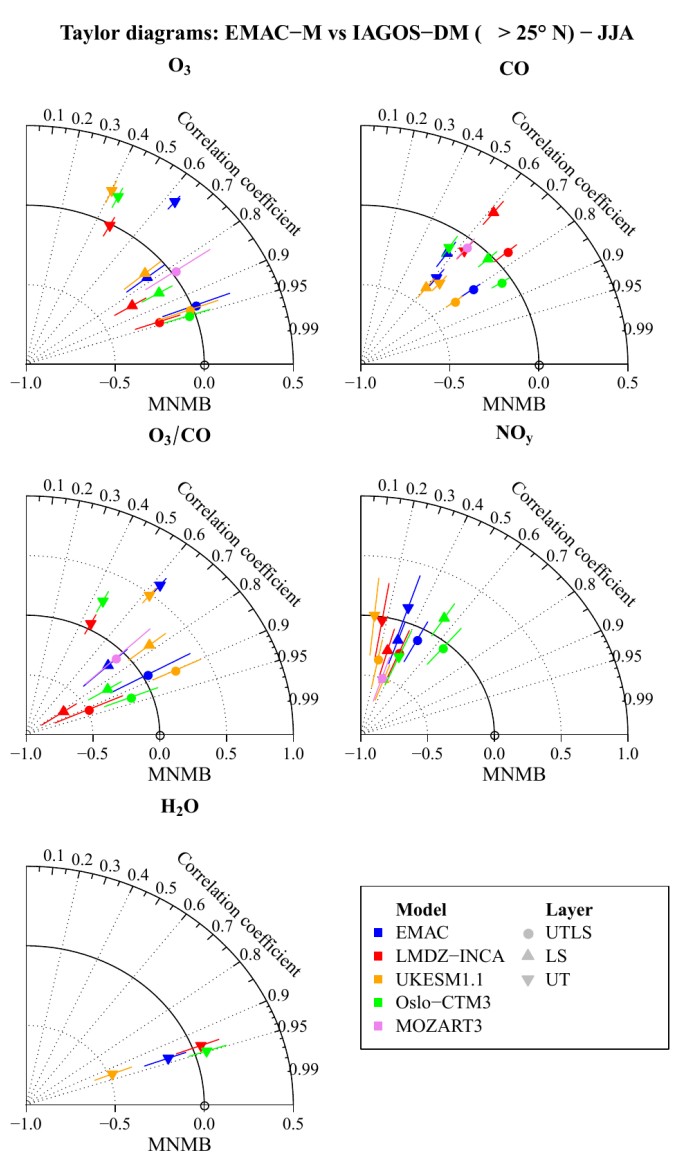

**Figure D3.** Same as Fig. 10 for boreal summer.




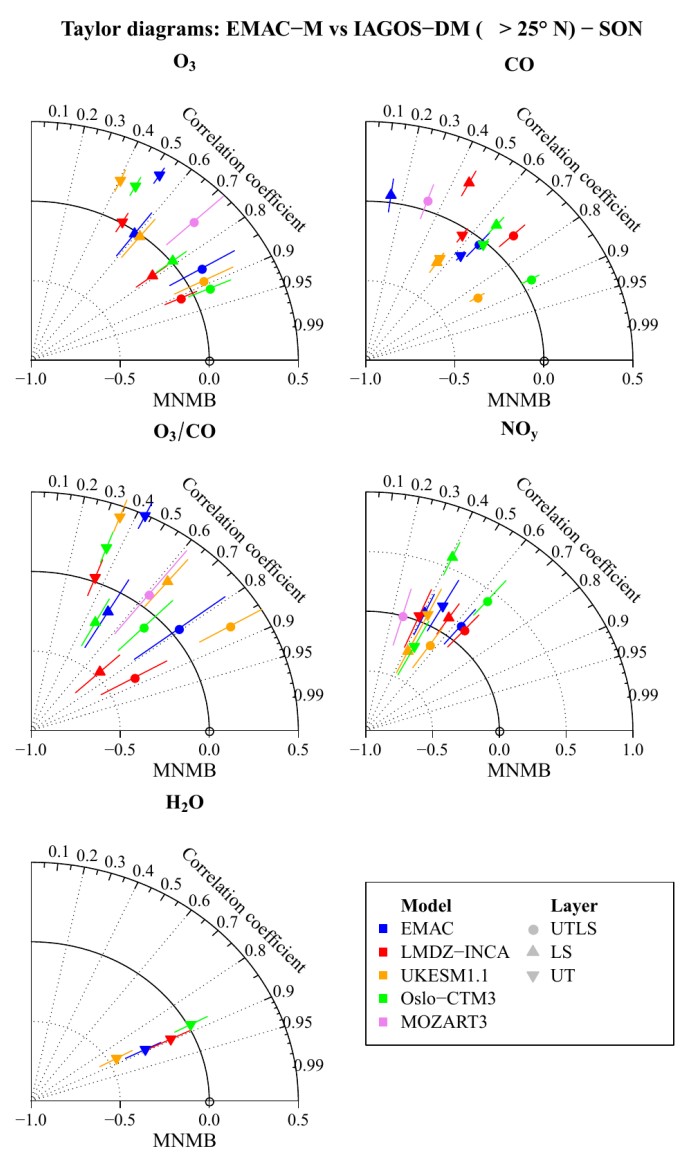

**Figure D4.** Same as Fig. 10 for boreal fall.



*Code and data availability.* The IAGOS data (IAGOS, 2022) are available at the IAGOS data portal (https://doi.org/10.25326/20) and more precisely, the time series data are found at https://doi.org/10.25326/06 (Boulanger et al., 2018). The Interpol-IAGOS software is available at https://doi.org/10.25326/81 (Cohen et al., 2020).

*Author contributions.* YC designed the study, and developed further the Interpol-IAGOS software. DH designed the modeling protocol shared by the five models. The simulations output were provided by SM, RT for EMAC, YC and DH for LMDZ-INCA, AS for MOZART3, MTL for OsloCMT3, and NB for UKESM1.1. The IAGOS data were provided by VT, AP, SR, UB, AZ and HZ. The paper was written by YC and reviewed, commented, edited and approved by all the authors.

*Acknowledgements.* The authors acknowledge the strong support of the European Commission, Airbus, and the Airlines (Lufthansa, Air-France, Austrian, Air Namibia, Cathay Pacific, Iberia and China Airlines so far) who carry the IAGOS-CORE equipment and perform the maintenance since 1994. In its last 10 years of operation, IAGOS-CORE has been funded by INSU–CNRS (France), Météo-France, Université Paul Sabatier (Toulouse, France) and Research Center Jülich (FZJ, Jülich, Germany). IAGOS has been additionally funded by the EU projects IAGOS-DS and IAGOS-ERI. The IAGOS-CORE database is supported by AERIS. Data are also available on the AERIS web site www.aeris-data.fr. The simulations were performed using HPC resources from GENCI (Grand Équipement National de Calcul Intensif) under the gen2201 project. We also wish to acknowledge our colleagues from the IAGOS teams in FZJ, LAERO, DLR and KIT for all the preparation of the IAGOS and CARIBIC data used in this study. Notably for their contribution to the IAGOS–Core $NO_y$ data, we thank their former PI Andreas Volz-Thomas, and Karin Thomas who was involved in the processing of the data.

*Financial support.* This research has been funded by the European Union Horizon 2020 research and innovation programme under the ACACIA (grant agreement no. 875036) project, and by the Direction Générale de l'Aviation Civile (DGAC) under the ClimAviation project. MTL acknowledges funding by the Research Council Norway (grant no. 300718 Aviate) and the resources from the National Infrastructure for High-Performance Computing and Data Storage in Norway (grant no. NN9188K).

*Competing interests.* At least one of the co-authors is a member of the editorial board of Atmospheric Chemistry and Physics.



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
