# Peer review of "Evaluation of $O_3$ , $H_2O$ , CO and $NO_y$ climatologies simulated by four global models in the upper troposphere–lower stratosphere with the IAGOS measurements"

_EGUsphere, 2024_

## Author Comment (AC1)

We kindly thank the referees for their relevant comments, suggestions, and corrections.

The responses are organized as follows: the reviewer's comment is in blue, our answers are in black, and the changes proposed for the revised manuscript are in italics (black for modified sentences, grey for unchanged sentences that have been copy-pasted here to remind the context). The numbering of the pages and lines corresponds to the preprint, not to the modified document. Last, we answer "Done." to all the comments that suggest a modification, that we completely agree with, and that do not require any clarification in this document.

Before addressing the comments, please note that we found an inconsistency in the text, and corrected it as follows:

P31, L456: *"Another limitation of this approach is that the mean altitude of the measurements changes with the latitude: as the tropopause altitude decreases with the latitude, the subtropics are more sampled in the UT than in the LS, and reversely, the high latitudes are more sampled in the LS than the UT"*

This has been replaced by:
*"Another limitation of this approach is that* **the tropopause altitude decreases with the latitude whereas the cruise altitude does not depend on latitude. Consequently,** *the subtropics are more sampled in the UT than in the LS, and reversely, the high latitudes are more sampled in the LS than the UT."*

**Comments from Referee 1**

This manuscript compares several chemical transport models to in-situ aircraft data from commercial aircraft. The manuscript is generally good, with appropriate methodology for the comparisons. It could be a bit clearer in some points, and is probably publishable with substantive but minor revisions. The current version has a lot of grammar mistakes. I tried to correct the ones that interfere with interpretation but it needs a careful edit.

First, we thank Referee 1 for the detailed review, which undoubtedly represents a serious amount of work and contributes significantly to the quality of the paper. We tried to address as many other grammar mistakes as we could (notably using the "Grammarly" AI tool to find them), and we hope the English writing is now correct.

A few general comments:

1. The paper needs to be a bit clearer about the fact that the models are actually all running as transport models, even those which are described as Chemistry-Climate Models. It's only in one line and the table where this is spelled out.

This information is now more visible, notably with the recently added justification at the end of Section 2.1 (Simulation setup). The former version:

*"The runs are nudged or forced by horizontal winds taken from a reanalysis."*

has been replaced by:

*"In order to assess the model's ability to simulate the mean UTLS composition, it is important to provide simulations with the most realistic transport conditions. This is why the runs from the CCMs are nudged by horizontal winds taken from a reanalysis, indicated in Table 1. Three models are CCMs, and are nudged with ERA-Interim (EMAC and LMDZ-INCA) or ERA5 (UKESM1.1), and two models are CTMs forced by ERA-Interim (MOZART3) or the ECMWF OpenIFS product (OsloCTM3)."*

2. I'm not sure why you don't get PV from the driving meteorology of the MOZART model to use like the other models.

The MOZART output had been submitted with a PV field but was not usable as some artefacts were found, and the generation of the PV variable from MOZART would have required rerunning the whole simulation. During the discussion phase however, we recalculated a PV field from ERA-Interim (as it is the reanalysis forcing transport in MOZART) in order to address this relevant comment, but we finally realized that the MOZART simulation setup was different than expected: the meteorology was found to be the same every year - the one from 2007 - which does not allow for a realistic assessment. Thus it is not consistent with the required conditions for the assessment against IAGOS. Most of the figures including MOZART have been removed from this paper (notably the figures that quantify a bias compared to IAGOS), though we kept some figures that we still judged relevant, at least to show the seasonal behaviour.

The manuscript has been modified as follows:

1/ Description of the experiment setup:

*Three models are CCMs and are nudged with ERA-Interim (EMAC and LMDZ-INCA) or ERA5 (UKESM1.1),* *and one model is a CTM forced by the ECMWF OpenIFS product (OsloCTM3), similar to ERA-Interim. The simulation from another CTM participating in the ACACIA project (MOZART3, forced by ERA-Interim) has been included to present the model behaviour during 1997--2007, but the dynamical field is a cyclic repetition of the year 2007, which removes the interannual variability of atmospheric transport and thus cannot be treated as a model assessment, except to comment on the seasonality.*

2/ In Fig. 9, the MOZART3 boxplots have been dashed, as in Fig. B4.

3/ Most MOZART3 skills have been removed from Table 5. The only ones that remain are relative to the seasonal cycle.

4/ The MOZART3 assessment skills have been removed from Figs. 10, and D1–D4, and the lines corresponding to MOZART3 zonal cross sections have been removed from Figs. 14, and 15. The legends have been modified accordingly.

5/ Most non-separated UTLS figures have been removed: Figs. A1, and C1. In Fig. 10, the skills in the non-separated UTLS have been removed from all the models (as the main justification was to keep a common metric to include MOZART3 into the intercomparison).

6/ Other modifications in the text:

- P14, L316–317, removal: *"The mean biases… in Fig. A1."*
- L317–321, as already described above in the document
- L322–325, removal: *"The biases in the non-separated UTLS… the tropopause altitude. With the separation between the UT and the LS,"*
- L325, removal: *"(now excepting MOZART3)"*
- P22, L413–416, removal of the text in black: *"The correlation coefficient is generally greater in the non-separated UTLS than in the UT or the LS, for the variables with an important vertical gradient (ozone, CO, $O_3$/CO, $NO_y$). As mentioned in Cohen et al. (2023), this difference suggests at least that the most important changes in the mean tropopause height are well represented, notably the meridian gradient as shown in Fig. C1. In the separated layers, ozone mixing ratio is generally more difficult to model in the UT than in the LS [...]"*
- P33, L491–493, removal: *"It is also the case with the IAGOS--MOZART3 transects (1997--2007) regarding CO above Africa in December--March, but the subsequent lessened amount of data added to the impossibility of filtering out the stratospheric air masses with the same criterion as the other models (PV) results in shorter meridian profiles, or even not shown."*
- P36, L522, removal of the text in black: *"In most cases, water vapour shows a small bias with LMDZ-INCA and OsloCTM3, and a dry **(moist)** bias with EMAC and UKESM1.1 **(MOZART3)**."*
- L530–531: removal of the text in black: *"and a peak in biomass burning contribution shifted 10° southward, as with the LMDZ-INCA **and MOZART3** models."*

3.  Also, not sure about the different years and how that is handled, particularly when looking at things like biomass burning or regions subject to a lot of interannual variability. I think the IAGOS data is the same years? But when you put models together it might be a problem

For each model, the months taken into account are the same between the two products IAGOS-DM-model and model-M. This way, the interannual variability (IAV) is taken into account for the comparison between a IAGOS-DM-model and a model-M product. Further information has been added for clarification, in the first paragraph of Section 2.4:

*"Concerning the model output, a daily mask is applied with respect to the IAGOS sampling Cohen et al. (2023) excluding the non-sampled daily gridpoints. This way, the subsequent monthly products are representative of the same gridpoints and the same days. **As in Cohen et al. (2023), their whole name is model--M (the --M suffix referring to the IAGOS mask), but except in this section where there can be confusion, we refer to it using simply the model name.** Last, the seasonal and annual climatologies are then derived from the monthly means with the same method and filtering as in Cohen et al. (2023). **For each model, the pair of products IAGOS-DM-model and model--M are thus representative of the same time period as well.**"*

However, the period can indeed differ between two distinct IAGOS-DM products. For example, the IAGOS-DM-Oslo and the IAGOS-DM-EMAC products are based on different periods (respectively: 2001 – 2017, and 1994 – 2017). So the comparison between two

models (in terms of skill) can be affected by IAV, with the shortest-term climatology (from OsloCTM3) potentially more affected. It was particularly true for CO in IAGOS-DM-MOZART3 climatologies, due to the important biomass burning IAV. This point was emphasized in this call for caution at the start of Section 3, L317–321 in the preprint:

*"We can notice the sampling differences between the climatologies from MOZART3 (1997–2007), OsloCTM3 (2001–2017) and the other three assessed models (1994–2017). Interannual variability is therefore likely to cause differences in the observed climatologies especially for MOZART3, which period only includes six years of frequent CO observations with IAGOS-CORE, a species with a great IAV due to biomass burning emissions (Voulgarakis et al., 2015)."*

Now that the MOZART3 skills have been removed from this study as they were not relevant anymore, we do not refer to the IAGOS-DM-MOZART3 product anymore, thus the remaining differences linked to IAV in the observations concern the periods of OsloCTM3 (2001–2017) and that of the other three models (1994–2017). However, it does not impact observed CO climatologies, as CO observations started in December 2001.

The call for caution cited above has thus been changed as follows:

*"We can notice the sampling differences between the climatologies from OsloCTM3 (2001–2017) and the other three assessed models (1994–2017). Interannual variability is therefore likely to cause **moderate** differences in the observed climatologies **for ozone and water vapour in OsloCTM3, which period is 25% shorter than for the other models. It does not apply to CO and NOy as the frequent IAGOS-CORE observations started in 2001.**"*

4. I think some discussion of the dynamics of the models are warranted. I know they are nudged or CTMs, but the CCMs nudged can have dynamics which deviate from the driving meteorology and the different analyses may have a biased tropopause. Also the nudging can introduce spurious transport where strong gradients exist in the UTLS if the model is trying to go to another state biased versus the driving meteorology. A more dynamical view might help here (maybe plotting the tropopause height from all the models as a start).

The tropopause in the models has been investigated using a new set of figures added to Supplementary Material. They represent the pressure difference between observations and the 2-PVU tropopause for each model, both in the LMS and the UT. It is displayed in maps for ozone measurements (with a 10 hPa resolution), and in boxplots for each species. For each species, the annual differences are hardly perceptible between the models. The annual median ranges between 45 and 48 hPa in the UT (44–47 hPa for $NO_y$), and between -75 and -80 hPa in the LS. For ozone during the four seasons, the median ranges are respectively [-84 – -91 hPa], [-78 – -84 hPa], [-57 – -61 hPa], and [-77 – -82 hPa] in the LS, and [42–45 hPa], [41–43 hPa], [48–50 hPa], and [48–51 hPa] in the UT.

As to the spurious transport that can emerge from nudging, it is possible that this bias has been limited by our methodology regarding:

- the UT and LS definitions, which are separated by a transition layer starting at the first level below the tropopause up to the 3 PVU isosurface, thus isolating further these two layers.

- the exclusion of the grid cells where measured ozone is contradictory with the modelled PV value (less than 60 ppb in the LS, and more than 140 ppb in the UT): it avoids cases where the wrong layer has been selected because the dynamical field was highly inconsistent with the observations.

The text has been modified as follows, in Sect. 2.4, at the end of the second paragraph:

*"In the framework of the CCMI modeling experiment, Orbe et al. (2020) reports that the nudging process tends to enhance the transport variability between the CCMs and can generate artificial transport in the regions of strong gradients. This issue might be partly addressed by our methodology, notably the definition for the layers that enhances the isolation between them, and the exclusion of the grid cells with an inconsistent PV value regarding ozone observations. The mean pressure differences between observations and the model 2-PVU tropopause shown in Supplementary Material (Figs. S1--S5) do not exhibit noticeable differences between the models: mostly, they are less than 5 hPa (except in winter and spring for ozone and $NO_y$), and are always less than 10 hPa. For each species and layer, the distance between the sampled grid cells and the tropopause does not vary enough across the models to play a significant role in the inter-model discrepancies, as there is no apparent correlation with the chemical tracers."*

Specific comments:

Page 2, L11: Mention the tropopause in the abstract? Is the problem with the tropospheric tracers a tropopause height issue? Or some kind of averaging issue (even across a day, or multiple days in a month)?

Details have been added as follows:

*"It does not exclude other factors as an underestimation of CO emissions, an underestimation of transport from the surface, or an overestimated CO oxidation by the hydroxyl radical (OH), but the mean distance between observations and tropopause do not vary substantially across the models, and the intermodel differences are not correlated with these chemical biases."*

Page 2, L23: Reference Gettelman et al 2011

Gettelman, A., P. Hoor, L. L. Pan, W. J. Randel, M. I. Hegglin, and T. Birner. 2011. "The Extratropical Upper Troposphere and Lower Stratosphere." Rev. Geophys. 49 (RG3003). https://doi.org/10.1029/2011RG000355.

Done.

Page 2, L33: contribute —> contributes significantly to NOx mixing ratios

Done.

Page 2, L34: which is not correct grammar here. Awkward phrasing.

"Which" has been replaced by "whose".

Also lessor—>lower

Done.

Page 3, L40: Please check your use of articles (the). Here it should be accurately UTLS behavior and impact of aviation NOx (remove the). Next sentence: Ozone production (no the), but the background is correct. Chemicals are usually not 'the'.

Thanks for this correction. The "the" articles before chemicals have been removed.

Page 3, L45: on OH quantities (no the)

Done.

Page 3,L55: … in the model results and helps (but does not make a list in the next line)

Done.

Page 4, L76: higher than cruise altitudes.

Done, and for the other occurrences as well.

Page 5, L101: models' skill

Done.

Page 5, L110: what is the typical vertical resolution of models in the UTLS? I see this in Table 1, refer to it here. What actual range of pressure levels are you sampling?

The information in L110 has been completed as follows:

"[...] and a vertical resolution of 20 hPa at cruise altitudes. Initially, the models' vertical resolution at cruise altitudes ranges between 15--20 hPa (EMAC and UKESM1.1) and 25--40 hPa (LMDZ--INCA)."

To the second question, the answer has been added to the end of the first paragraph, Section 2.3:

"*The period we are analyzing spreads from Aug. 1994 until Dec. 2017.* **The most sampled altitudes are between 180 and 310 hPa, and the vertical distribution of sampling varies geographically.**"

Page 7, L135: I'm unclear the difference between EMAC, MESSy and MECCA. Please clarify. Which one is the modified ECHAM5 atmospheric model? Is one the chemistry module?

MESSy represents the modular software structure relying on submodels. The physics-related submodels have been mostly derived from ECHAM5 base model, and MECCA is a chemistry submodule for solving atmospheric chemistry equations.

The paragraph has been modified as follows, also accounting for the question on nudging:

*"The ECHAM/MESSy Atmospheric Chemistry (EMAC) model is a numerical chemistry and climate simulation system that includes sub-models describing tropospheric and middle atmosphere processes and their interaction with oceans, land, and human influences (Jöckel et al., 2010).* **It uses the second version of the Modular Earth Submodel System (MESSy2) to link multi-institutional computer codes. As described in Jöckel et al. (2016), MESSy is a software package providing a framework for a standardized, bottom-up implementation of Earth system models with flexible complexity (Modular Earth Submodel System). The core atmospheric model is the 5$^{th}$ generation European Centre Hamburg general circulation model (ECHAM5, Roeckner et al., 2006).** *The physics subroutines of the original ECHAM code have been modularized and reimplemented as MESSy submodels and have continuously been further developed. Only the spectral transform core, the flux-form semi-Lagrangian large-scale advection scheme, and the nudging routines for Newtonian relaxation remain from ECHAM. For the present study, we applied EMAC (MESSy version 2.55.2) in the T42L90MA-resolution, i.e. with a spherical truncation of T42 (corresponding to a quadratic Gaussian grid of approximately 2.8 by 2.8 degrees in latitude and longitude) with 90 vertical hybrid pressure levels up to 0.01 hPa.* **In ECHAM5, the nudging applies to vorticity, temperature, logarithm of the surface pressure, and divergence with a relaxation time being 6 h, 24 h, 24 h, and 48 h respectively.** *Using a so-called quasi chemistry--transport mode (QCTM, Deckert et al., 2011) enables binary identical simulations with respect to atmospheric dynamics, and perturbations in chemistry can be detected with a high signal-to-noise ratio. The applied model setup comprised the Module Efficiently Calculating the Chemistry of the Atmosphere (MECCA) used for tropospheric and stratospheric chemistry calculations with the possibility of extending to the mesosphere and oceanic chemistry (Sander et al., 2019). Reaction mechanisms include ozone, methane, $HO_x$, $NO_x$, NMHCs, halogens, and sulfur chemistry. Radiative transfer calculations are performed using the submodel RAD (Dietmuller et al., 2016)."*

Page 10, L216: which year? Most other models are running 2014-2018 at a minimum right?

This sentence has been rephrased as follows:

*"For ACACIA,* **the output is made of a succession of one-year simulations, each one with 6 months spin-up."**

Most other models are running 2014-2018 at a minimum right?

We realize that the last introduction paragraph was not clear, notably with the phrase L93–94 (P4) which can lead to confusion:

*"long-term simulations from five [...] models, involved in a multi-model experiment in the framework of the ACACIA [...] project. This experiment focuses on the present-day (2014--2018) and future (2050) impact of aircraft $NO_x$ [...] on climate".*

The article "This" did not refer to the current study, but more generally to the whole multi-model experiment that includes several runs from each model. We tried to give an overview of the multi-model experiment where the current study belongs.

We reorganized the introduction as follows:

- We added an introduction to the ACACIA project at the end of the paragraph starting with "Modelling UTLS behaviour accurately is an important step…" and "Assessing the models' abilities in reproducing…", as follows:

  *"In the framework of the ACACIA (Advancing the Science for Aviation and Climate) European Union project which focuses on the non-$CO_2$ effects from subsonic aviation on climate, a multi-model experiment has been performed using a set of runs from five state-of-the-art CCMs or CTMs. This modelling experiment aims to investigate the present-day and future impact of aircraft $NO_x$ and aerosol emissions on the atmospheric composition and therefore on climate, and consists of the analysis of runs with and without aircraft emissions, as presented in companion papers (Cohen et al., in prep; Staniaszek et al., in prep.; Bellouin et al., in prep.). "*

- The former paragraph concerned by the current question (P10, L216) has been modified as follows:

  *"The current study aims to extend the former assessment to the climatologies from long-term simulations from four CCMs/CTMs involved in the ACACIA project. While a companion paper focuses on the present-day sensitivity of the modeled atmospheric composition to aviation emissions (Cohen et al., in prep.), the current paper consists of assessing (bi-)decadal climatologies derived from the main run of every model against the IAGOS data, in the UT and the LS separately, as done in Cohen et al. (2023) for the LMDZ--INCA model."*

Page 11, L261: This doesn't quite overlap most of the model results (until 2018). Are you going to compare the models to the correct years? Some models are nudged and some are not. Is that going to matter? It's not clear from the description exactly how you did this, and how you will do comparisons across models if they are different years.

As indicated in the simulation setup description (P5, L117), the historical runs cover the period 1994–2017 for EMAC, LMDZ–INCA, and UKESM1.1, the period 2001–2017 for OsloCTM3, and 1997–2007 for MOZART3. It has been clarified as follows:

*"The period we are analyzing spreads from Aug. 1994 until Dec. 2017, hence the 1994--2017 (or 2001--2017) time period covered by the models' run."*

For the method used to assess the models' runs, the following sentence has been added at the end of the paragraph:

*"The methodology used for the models' assessment using the IAGOS data is described in Section 2.4."*

In the end, the paragraph ends with these phrases:

*"The period we are analyzing spreads from Aug. 1994 until Dec. 2017, hence the 1994--2017 (or 2001--2017) time period covered by the models' run. The most sampled altitudes are between 180 and 310 hPa, and the vertical distribution of sampling varies*

*geographically. The methodology used for the models' assessment using the IAGOS data is described in Section 2.4."*

Page 12, Table 2: Can you indicate dates for IAGOS-CARIBIC and IAGOS-CORE? Since the merge in 2008, which measurements have continued?

A last column has been added to Table 2, indicating the time period. The legend has been completed as follows:

*"Table 2. Characteristics of the IAGOS instruments measuring ozone, CO, water vapour, and $NO_y$. The last column shows the time period covered by the measurements. The periods ending with a hyphen mean that the measurements are still ongoing."*

In the end, only $NO_y$ from IAGOS-CORE has an ending date, while all the other species are still measured.

Page 12, L282: Are all the models using nudged meteorology so they can get the right comparisons? This is confusing because they are, but several of the models are nudged CCMs.

Three of the models are using a nudged meteorology (EMAC, LMDZ–INCA, and UKESM1.1), and the OsloCTM3 model is a CTM.

We use nudged meteorology in order to be able to compare to the measurements, and to have the meteorology close to the measurements conditions mostly to represent long-range transport properly. We need to stress at the ultimate objective, i.e. to investigate the climate impact of aviation and therefore CCMs are needed. In order to evaluate these CCMs and compare to observations, we hence have to apply nudging procedures for the winds.

Also: you are sampling daily and then dividing by a daily averaged PV? How much uncertainty will this introduce if you are off by a few grid boxes? (At 10m/s, that's 900km/day, so could be a few grid boxes off).

We apologize as there might be some confusion. The software does not divide any variable by PV.

In case Referee 1 asks for confirmation that the daily grid boxes are classified in one or another layer depending on their PV daily value, the answer is yes. But as explained in the second paragraph of Section 2.4 (L 285–289), a grid box in the UT or in the LS is validated only if the daily average derived from ozone observations is considered as consistent with this layer.

In case the model's PV is off such that the current grid box is classified in the wrong layer, then it is detected with ozone observations that are inconsistent with the "chosen" layer. Then the concerned daily grid box is not taken into account, as the aim is not to assess the tropopause location but the UT and LS chemical compositions.

Page 13, L289: so you are filtering out stratospheric intrusions? How often does the filtering kick in? Same for models and IAGOS data?

If stratospheric intrusions are resolved, then the PV inside the intrusion is higher. If it is higher than 3 PVU, then it is considered as lower-stratospheric, without consideration of being a stratospheric intrusion or in the core of the LMS.

Page 13, L290: Why don't you just get a PV field from the ERAI data? That seems easy to do. They probably even have a PV field already.

To address this highly relevant comment, we recalculated a PV field from ERAI, but as explained above in this document, we realized that the MOZART3 dynamical field did not meet the protocol described in the simulation setup. As the inclusion of MOZART3 into the comparison was the main purpose for the assessment in the non-separated UTLS, the latter has also been removed from the study (except Fig. 9, where MOZART3 has been dashed for caution), in order to lower the high number of figures.

Page 13, L294: define explicitly how the MNMB and FGE are calculated. It is not obvious.

It is now defined in the Supplementary Material as equations S1–S4, same as in Cohen et al. (2023):

*"As in Cohen et al. (2023), the chosen metrics are the modified normalized mean bias (MNMB) with the fractional gross error (FGE), and Pearson's correlation coefficient, **all defined in Supplementary Material (Eqs. S1– S4)."***

Page 13, L296: What is the IAGOS-DM product? So you use different years for different models?

It is defined in section 2.4, but it lacks clarity as it is not said explicitly. We thus clarified as follows:

- Section 2.4, first paragraph: *" [...] the -DM suffix referring to the distribution onto the model grid. **For further simplicity in this study, we refer to it as IAGOS--model, and the IAGOS-DM-model products in their ensemble are called "IAGOS--DM products".***
- Section 3.1, first paragraph (corresponding to P13, L296 in the preprint): "*[...] relative to a different gridded IAGOS product* (called hereafter IAGOS--DM, see Section 2.4)"

Page 13 L296: Does your methodology of using monthly data provide the same discrimination as if you used daily or 6 hourly data? I worry that the monthly averaging will smear out the dynamics and the chemistry, and can hide biases in the UT and LS when mixed? Can you show you get the same answers as using daily data with at least one model? I get that you are averaging daily to monthly, but the altitudes will get quite smeared out in the UT and LS.

We are using a daily resolution to discriminate the UT/LS grid cells. The monthly averages are then calculated separately in the UT and in the LS. For a monthly average, a given grid cell can have both an upper-tropospheric mean value and a lower-stratospheric mean value if it has been in both layers during the month.

As there might have been a lack of clarity in our explanations about the methodology (as pointed out in this document), we added further precision in the first paragraph of Section 2.4:

*"Concerning the model output, a daily mask is applied with respect to the IAGOS sampling (Cohen et al., 2023), excluding the non-sampled daily gridpoints. This way, the subsequent monthly products are representative of the same gridpoints and the same days. As in Cohen et al. (2023), their whole name is model--M (the --M suffix referring to the IAGOS mask), but except in this section where there can be confusion, we refer to it simply using the model name.* **The monthly average is calculated for each layer separately. It implies that for each grid cell that has been in both UT and LS during the month, a monthly average is calculated for both layers.** *Last, the seasonal and annual climatologies are then derived from the monthly means with the same method and filtering as in Cohen et al. (2023). For each model, the couple of products IAGOS-DM-model and model--M are thus representative of the same time period as well."*

Page 13, L306: I've lost which region is which. Maybe you need a table here to delineate the regions and time periods.

We agree that there is a need to synthesize the regions' characteristics. A new table has been set, based on Table 1 in Cohen et al. (2023).

Page 13, L309: How are the UT and LS separated? Are all these nudged simulations? I'm assuming they have to be since you are sampling models at the IAGOS data times right? But again, this information is buried in one comment in the methodology. You might need to repeat it here.

As suggested by Referee 1, a reminder has been added to the paragraph, as follows:

*"[...] and finally move into the (sub-)tropical UT characterization.* **It is worth reminding that the principal criterion for the UT and LS definition beyond +/- 25° N is the 2-PVU isosurface from the nudged CCMs/forced CTM, and that the tropical UT comprises every sampled grid cell above 300 hPa (as our sampling does not reach the tropical tropopause layer)."**

Page 13, L314: The different products are because of different time periods right?

Yes, and because of different dynamical fields too, but the effects from the different time periods are more visible. It is now clarified as follows:

*"A climatology is also shown for one of the gridded IAGOS products (called hereafter IAGOS--DM products, see Section 2.4) in order to provide a view of the expected features, but each bias remains relative to a different IAGOS--DM product* **notably because of the different time periods.** *Since the IAGOS--DM climatologies are relatively similar through the simulations with the same duration (not shown), [...]"*

Page 14, L316: That is what is displayed on the far left? Please be explicit.

It has been clarified as follows:

*"Since the IAGOS--DM climatologies are relatively similar through the simulations with the same duration (not shown), we chose only one of the IAGOS--DM climatologies with the longest time period (by default IAGOS--EMAC,* **i.e. the gridded IAGOS product on EMAC model's grid, as explained in Section 2.4).*"**

Page 14, L325: Are you going to show the tropopause altitude/temperature? Seems like that's important here. What about using tropopause relative coordinates? Would that reduce the biases? I can see how different analyses may have different tropopause heights, and the nudging may also introduce different tropopause heights.

The main purpose of showing biases in the non-separated UTLS was to allow comparisons for models that did not diagnose the tropopause altitude. We agree that if this variable is available, then not using it becomes hard to justify. For a given layer, the pressure differences are relatively low across the models. Generally, it is not higher than 5 hPa, except in winter and spring in the LS, for $NO_y$ and ozone, but it does not reach 10 hPa (~ 200 m). It could still be problematic for water vapour in the LMS as the vertical gradient from the tropopause is particularly high, including two orders of magnitude (Zahn et al., 2014), but there is no apparent link between Figs. 5–8 and Figs. S2–S5. For instance, ozone (Fig. 5) is higher in the UT with EMAC and lower with LMDZ-INCA, but the sampled grid cells are closer to the model tropopause (Fig. S2) for LMDZ-INCA. In the LS, the sampled grid cells are closer to the tropopause for EMAC and UKESM, and yet, they do not minimize ozone.

The comparison with Figs. S2–S5 is added to the manuscript as follows:

*"This issue might be partly addressed by our methodology, notably the definition for the layers that enhances the isolation between them, and the exclusion of the grid cells with an inconsistent PV value regarding ozone observations. The mean pressure differences between observations and the model 2-PVU tropopause shown in Supplementary Material (Figs. S1--S5) do not exhibit noticeable differences between the models: mostly, they are less than 5 hPa (except in winter and spring for ozone and $NO_y$), and are always less than 10 hPa. For each species and layer, the distance between the sampled grid cells and the tropopause does not vary enough across the models to play a significant role in the inter-model discrepancies, as there is no apparent correlation with the chemical tracers."*

Page 14, L332: But wouldn't the bias be the opposite across the tropopause if there were two much STE? I.e. too much O3 in the UT and too little in the LS, with the opposite for CO? Your statement implies the same sign bias. Please clarify.

It is true that too much STE would lead to opposite biases in the two areas, but it was not what we suggested. We rather suggested a bias in one area that extended to the other area through STE.

We clarified this hypothesis as follows:

*"We can notice that the models showing a more positive (negative) bias in ozone (CO) in the low-latitude LS have the same tendency in the tropical UT,* **possibly** *as an impact of isentropic cross-tropopause exchanges* **that can extend biases to the adjacent layer."***

Page 14, L336: That means highs and lows aren't large enough? I.e. too little variability?

Yes. We added this clarification to this sentence:

*"The four models generally underestimate the magnitude of these geographical extrema, showing a too small variability."*

Done.

Done.

Page 19, L370: To what extend does the driving meteorology (e.g. IFS or ERAI/5) have a realistic Brewer-Dobson circulation? Similar to comment about tropopause height above.

Ploeger et al. (2021) used $CO_2$ measurements from 5 aircraft campaigns, and diagnosed a slow-biased Brewer-Dobson circulation (BDC) in ERA5 in the NH lower stratosphere (350–480 K). The ERA-I age of air tends to be underestimated in this area, but it remains generally well inside the observation uncertainties and thus fits better with the observations than ERA5. Li et al. (2022) also found a longer age of air in the northern midlatitude LS in ERA5 than in ERA-I, especially in boreal winter. In our results though, it appears that it is not the main factor that influences ozone in the LS, as ozone in UKESM1.1 is well inside the model range, in every season. Also, it does not explain the low $HNO_3$ values in this model, as it is now clarified in the manuscript (P23, paragraph L412–420 in the new preprint):

*"For UKESM1.1 (OsloCTM3), the lower (higher) $HNO_3$ values in the LS (Fig. B3) might thus be due to an underestimation (overestimation) of $N_2O$ flux into the stratosphere, to overestimated (underestimated) $N_2O$ lifetime in the stratosphere, or to underestimated (overestimated) $HNO_3$ lifetime against stratospheric aerosol uptake. The latter is a possible contributor for OsloCTM3, as its mass density of particular sulfate and nitrate is lower by 10 % than in the LMDZ-INCA simulation. Lower $NO_y$ and $HNO_3$ mixing ratios in UKESM1.1 are unlikely due to the different representation of the Brewer--Dobson circulation in the reanalyses, as the mean age of air in the northern LS is longer in ERA5 than in ERA-Interim (Ploeger et al., 2021; Li et al., 2022), which would tend to convert further $N_2O$ into $HNO_3$ in UKESM1.1. In the end, considering both ozone and $NO_y$ in the LS, the similarities between observations and models, notably during spring, are encouraging regarding the stratospheric chemistry and diabatic transport for all the models."*

The following references have been added:

- Ploeger, F., Diallo, M., Charlesworth, E., Konopka, P., Legras, B., Laube, J. C., Grooß, J.-U., Günther, G., Engel, A., and Riese, M.: The stratospheric Brewer–Dobson circulation inferred from age of air in the ERA5 reanalysis, Atmos. Chem. Phys., 21, 8393–8412, https://doi.org/10.5194/acp-21-8393-2021, 2021.
- Li, Y., Dhomse, S. S., Chipperfield, M. P., Feng, W., Chrysanthou, A., Xia, Y., and Guo, D.: Effects of reanalysis forcing fields on ozone trends and age of air from a

chemical transport model, Atmos. Chem. Phys., 22, 10635–10656, https://doi.org/10.5194/acp-22-10635-2022, 2022.

Page 19, L384: Are the representations of the stratospheric circulation the same in the different simulations? How much could they differ?

Renewing the inter-model comparison using other variables (like mean age of air) would be relevant to understand more accurately the cause of the model differences, but it represents too much additional work, given that they are not available in the initial model output that have been shared for the analysis. However, it is worth being mentioned as a perspective, as follows:

*"Concerning the impact from the Brewer–Dobson circulation on the LS, a better understanding of the models' biases could be brought by assessing the dynamical behaviour exclusively, hence adding other variables like stratospheric age of air would be relevant for a more complete model evaluation, or the zonal momentum (Diallo et al., 2021)."*

Page 22, L416: Show the tropopause height? Also: meridian —> meridional

As written above in this document, the comments on the non-separated UTLS have been removed (with the "meridian" adjective as well). More precisely, the following text in black has been removed:

*"The correlation coefficient is generally greater in the non-separated UTLS than in the UT or the LS, for the variables with an important vertical gradient (ozone, CO, $O_3$/CO, $NO_y$). As mentioned in Cohen et al. (2023), this difference suggests at least that the most important changes in the mean tropopause height are well represented, notably the meridian gradient as shown in Fig. C1. In the separated layers,* ozone mixing ratio is generally more difficult to model in the UT than in the LS [...]*"*

Page 26, L419: Again, please provide the formula for FGE and MNMB. The MNMB I can guess, but do not know what you mean by FGE.

Done, as described above. The equations added to the manuscript also highlight the meaning of the case FGE = | MNMB |, as it is used in the analysis of the results.

Page 26, L424: Can you separate dynamics from chemistry here? Perhaps with another tracer (e.g. H2O).

A comparison between water vapour and CO was made a little bit after, L425. It has been completed as follows:

*"In the UT, ozone and CO biases tend to be respectively positive and negative. This antagonism can be linked with overestimated cross-tropopause exchanges and/or an overestimated photochemical activity, thus with more ozone production causing more CO destruction.*

*In the UT as well, both surface tracers (CO and $H_2O$) show good correlations [...]. The skill difference between the two species can be explained either by uncertainties in CO*

*emissions in each region, or an underestimation of the detrainment altitude from pyroconvection* **consistently with the negative biases in CO in the UT.”**

Also, as ozone is not the only OH precursor (nitrogen oxide is more important with the conversion of $HO_2$ into OH), we replaced the argument (same quote above):

“*thus with more ozone production causing more CO destruction.*“

by:

“*thus with more ozone production* **and** *more CO destruction.*“

 So I'm assuming that EMAC and UKESM are free running models then? No nudging? That seems to be your description but then you are nudging them... what is the nudging timescale? Could the tropopause vary from the driving analysis?

To the first question, only the horizontal winds are nudged (and sea-surface temperatures are forced) for LMDZ–INCA and UKESM1.1. The other meteorological variables mentioned here (spatial distribution of lightning emissions, and cloudiness representation) are calculated by the GCMs.

To the second question, the relaxation time is 3.6 h for LMDZ–INCA, and 6 h for UKESM1.1. Concerning ECHAM5 used in EMAC, the nudging applies to vorticity, temperature, logarithm of the surface pressure, and divergence with a relaxation time being 6 h, 24 h, 24 h, and 48 h respectively. All these points are now indicated in each model's presentation.

To the third question, as explained above in this document, we do not find important inter-model differences in the tropopause altitude, and the differences in the chemical tracers do not correlate with these altitude differences.

 What is with UKESM in Figure 11? Why is it an outlier?

It is true that looking at the analysis, one can expect a comment on UKESM. As this feature has been commented on for the boxplots in Fig. 5 (L362–364):

*“The lower values in the two tropospheric tracers ($H_2O$ and CO) in both UT and LS with UKESM1.1 suggest an underestimation in the upward fluxes from the surface up to the UT, which favours an underestimation in the LS too.”*

the following text has thus been added to L446 as a reminder:

“*Concerning EMAC, [...] given that the latter are probably overestimated.* **Concerning UKESM1.1, it is worth reminding that the low mixing ratios in $H_2O$ and CO in both layers (see Figs. 5–6) suggest an underestimation in the upward fluxes from the surface up to the UT.”**

 I read it that the gradient for LS O3 is okay except for the sub-tropics, while for LS NOy the subtropics are slightly less biased.

It is true that the phrasing *"The situation is similar for NO$_y$"* is vague. For more clarity, we rephrased it as follows: *"The northward gradient is also visible for NO$_y$ with an underestimated regression slope as well, ..."*.

Page 28, L452: Not sure what this last sentence 'smaller scales are hardly captured' means. How do you know that from this scatterplot?

This sentence has been clarified:

*"Contrary to ozone, this is characterized by poor correlations inside each zonal band. It suggests that NO$_y$ correlation in the LS is mostly due to the northward gradient and that the **zonal distribution is** hardly captured by the models for this variable."*

where *"zonal distribution"* replaces *"smaller scales"*.

Page 32, L461: Do these models have more realistic stability? Presumably the CTMs do, but what about the CCMs? They might not interact well with the nudging?

If the comparison suggested by Referee 1 concerns the CCMVal-2 models and our study, it is hard to compare numerically as we do not use the same definitions for the UT/LS, and as the observation data sets (satellite and aircraft campaigns) are different from this study.

Also, the simulations assessed in Hegglin et al. (2010) are not nudged. This information has been added L461:

*"First, most of the CCMVal-2 **free-running** models assessed in Hegglin et al. (2010) [...]"*

Consequently, and according to Referee 1's comments on nudging, we added the following sentence:

*"Still, it does not seem to be the most controlling factor for water vapour, as the EMAC model is one of the most resolved models but has the weakest water vapour vertical gradient, **unless the nudging makes this inter-model hierarchy less evident.**"*

Page 32, L470: it would be useful to look at tropopause relative gradients to try to understand the model UTLS separation.

We totally agree with Referee 1 on the importance of a vertical profile with tropopause coordinates, as it would bring more detailed information than the simple two layers used in our study (though they are defined relatively to the tropopause as well).

But for this purpose, we would need to subdivide the UTLS into ~10 relative altitude bins or more (for 4 models), which would make the software much slower, and the output quite long to generate. The assessment using the two layers UT and LMS is a first step nevertheless, as it reveals relevant model characteristics, hence a well-resolved calculation for vertical gradients is beyond the scope of this study. Still, this suggestion has been mentioned in the perspectives:

*"In the extra-tropics, the choice of more accurate dynamical coordinates as the equivalent latitudes (involving both potential temperature and PV) or the jet-related tropopause*

*\citep{millan2024} will probably improve the model assessment*, as well as vertical profiles with tropopause-relative coordinates.”

Page 32, L481: You don't really comment much on the O3 and NOy UT scatterplots. Think this needs another paragraph.

Thanks to Referee 1 for pointing this out, we missed an interesting feature for ozone in the UT. The following paragraph has been added after the paragraph on the LS scatterplots:

“*Concerning the upper troposphere in Fig. A3, the observed ozone climatology does not show any latitude gradient, with yearly means ranging between ~50 and 80 ppb independently of the latitude bin, except for some subtropical locations that are poorer in ozone and can reach ~35 ppb. The EMAC model reproduces this feature relatively well, though with a systematic overestimation. The other models do not differentiate the northernmost two bins (45--55° N and 55--65° N), but they tend to make a distinction between 25--35° N, 35--45° N, and 45--65° N. LMDZ--INCA and OsloCTM3 tend to simulate a northward gradient. As it is a characteristic of the LS, it would be consistent with an overestimated stratosphere–troposphere transport. Concerning UKESM1.1, a significant part of subtropical ozone values are higher than in all the high latitudes. It might be a consequence of the lower-stratospheric biases into the UT through cross-tropopause exchanges, as in the LS, subtropical ozone is particularly overestimated and high-latitude ozone is less abundant than in the other models. For $NO_y$ in the UT shown in Fig. A5, we also notice that the models simulate more $NO_y$ in the high latitudes and less in the subtropics as in the LS, which is not consistent with the observations and suggests overestimated cross-tropopause exchanges as well.*”

Done.

The phrasing here was too brief, thanks to Referee 1 for pointing this out. Most of the paragraph has been rephrased:

“*In a second step, we present an overview of model skill.* **The mean pressure is represented as well, in order to identify the changes in the observed variable that can be associated with changes in the sampling mean altitude. This case occurs at the edge of the sampled regions, notably for $NO_y$ during December–February above South America, and during June–October above Africa. The IAGOS–OsloCTM3 profiles are represented as well, as their sampling period is shorter than the other three models (2001–2017 instead of 1995–2017), which causes differences in the IAGOS–OsloCTM3 transects in ozone (July–August in South America) and water vapour (June–October above Africa).**”

Page 33, L491: This last sentence needs some significant grammar corrections. 'Lessened', 'impossibility to' and 'meridian' are not correct.

This sentence has been removed as it was about MOZART3 zonal cross sections, which are not represented anymore.

Done.

The sentence has been clarified:

*"During the meridional transport in the upper branch of the strongest Hadley cell, CO accumulates* **in the areas where zonal wind shear is greater** *(Sauvage et al., 2007; Lannuque et al., 2021)* **as for NO$_y$,** *reaching a maximum at approximately 10° away from the ITCZ."*

Done.

In this sentence, we meant that Asian Summer Monsoon is the strongest convective system (stronger than the others), thus we chose to keep the singular form. We rather rephrased as *"representation of the strongest convective system"* instead of *"representation of the most convective system"*.

Page 33, L511: about representation of convective systems

We rather chose to rephrase as *"representation of strongest convective systems"*.

Page 36, L517: most? Do you mean most (delete the article then) or moist? Not sure what 'favored' means. Pick a different verb to clarify please.

We really meant "the most convective", but "strongest convective systems" might be clearer. We rephrased it. We also replaced "favored" by "improved".

Page 36, L520: peaks' intensity (the possessive I think is needed here).

Done.

Page 36, L5235: also in the southward CO shift? What is this referring to? biomass burning? Awkward phrasing.

We rephrased it as follows:

*"southward shift of CO peak"*

Page 36, L539: Are the models being compared to the same years of the observations? Could interannual variability being different explain the difference? Or are you matching years so it is missing biomass burning? I think different models have different years? That seems problematic.

It is true, and this is why we represent the IAGOS zonal cross sections for different models as well. The comment on the shorter period for OsloCTM3 (P33, L488–491) has been rephrased as follows:

*"The IAGOS--OsloCTM3 profiles are represented as well, as their sampling period is shorter than the other three models (2001--2017 instead of 1995--2017),* **which causes important differences only in the IAGOS--OsloCTM3 transects in ozone during July--August in South America, and in water vapour during June--October above Africa."**

Without MOZART3, it is not the case anymore with CO and $NO_y$, as the four remaining models have their simulation beginning before the start of the IAGOS–CORE measurements (April and December 2001, for $NO_y$ and CO respectively), and ending at the same time (2017).

Page 37, L532: lessened —> lower

(L547) Done.

Page 37, L549: do you mean that overestimating of lightning is a known bias in LMDZ-INCA? Reference?

The reference has been added, thanks to Referee 1 for reminding this need: Hauglustaine et al. (2004).

Page 37, L573: you don't mean directly comparable across models right? Just that for each model you compare DM and M? Clarify.

We confirm Referee 1's interpretation. We rephrased the sentence as follows:

*"For a given model, the subsequent IAGOS and model products are called IAGOS-DM-model and model-M respectively, the –DM and –M suffixes referring to the distribution onto the model grid, and to the IAGOS mask respectively. This way, each model product is directly comparable to the corresponding IAGOS-DM-model product."*

Page 38, L576: skills—>skill

Done.

Page 38, L577: Do the models have the right representation of the Extratropical Tropopause Layer? Why would they overestimate mixing? You might need to comment on this.

As our results are not sufficient to explain the reasons for our diagnostics, we can only provide hypotheses.

*"In the northern mid-latitudes, the results suggest that most models tend to overestimate the cross-tropopause mixing,* **which might be linked to a too diffusive extratropical transition layer."**

Page 40, L609: I think you need a bit more dynamical assessment. How much of the bias or extra cross tropopause transport assumed could be just your analysis method smearing out the tropopause? Is the tropopause height okay in the models?

Every measurement corresponding to a PV less than 2 PVU is considered upper-tropospheric, and more than 3 PVU for the lowermost stratosphere. A PV field is calculated by each model, nudged toward (or forced with) a reanalysis. As reminded by Referee 1, the PV field can be biased in one or several models, which can shift the tropopause altitude. Our methodology includes an additional filtering process based on observed ozone in order to limit the impact of PV biases, but it does not exclude smaller PV biases if ozone mixing ratio does not fall out of the expected range. It is only possible for us to compare the mean tropopause altitude across the models, which shows only small changes, at least on the seasonal timescale. As stated before in this document, we did not find any evidence of an influence between the tropopause altitude and the model biases.

---

## Author Comment (AC2)

We kindly thank the referees for their relevant comments, suggestions, and corrections.

The responses are organized as follows: the reviewer's comment is in blue, our answers are in black, and the changes proposed for the revised manuscript are in italics (black for modified sentences, grey for unchanged sentences that have been copy-pasted here to remind the context). The numbering of the pages and lines corresponds to the preprint, not to the modified document. Last, we answer "Done." to all the comments that suggest a modification, that we completely agree with, and that do not require any clarification in this document.

Before addressing the comments, please note that we found an inconsistency in the text, and corrected it as follows:

P31, L456: *"Another limitation of this approach is that the mean altitude of the measurements changes with the latitude: as the tropopause altitude decreases with the latitude, the subtropics are more sampled in the UT than in the LS, and reversely, the high latitudes are more sampled in the LS than the UT"*

This has been replaced by:
*"Another limitation of this approach is that* **the tropopause altitude decreases with the latitude whereas the cruise altitude does not depend on latitude. Consequently,** *the subtropics are more sampled in the UT than in the LS, and reversely, the high latitudes are more sampled in the LS than the UT."*

**Comments from Referee 2**

Cohen et al. processed long-term IAGOS measurements to generate climatologies of CO, O3, H2O, and NOy in the UTLS region, and then utilized these observations to evaluate five different chemical transport/chemistry-climate models. They also highlighted dynamic features observed in the IAGOS data, such as the seasonal shifts of the ITCZ above Africa. The manuscript is quite comprehensive, perhaps containing more results than can be effectively presented in a single paper. I hope the authors could address the following comments before being published in ACP.

Major comments: The manuscript is challenging to follow from beginning to end, as each section feels somewhat disconnected. For example, while the introduction emphasizes assessing the impact of aviation emissions, the paper itself does not present results specifically related to aviation emissions. Additionally, Sections 3.1 and 3.2 both discuss biases due to cross-tropopause transport, yet they are presented separately. Figures 10 and 11-13 seem redundant. The manuscript could be significantly improved by simplifying the introduction and focusing on presenting the most important results.

We kindly thank Referee 2 for the relevant suggestions, helpful for the clarity of the paper.

Figures 11–13 have been moved into the Appendix, but we keep their analysis in the manuscript. Their additional information is relevant, as they provide an understanding of the metrics shown in Figure 10, and meridional information as well.

According to Referee 1's comments too, several modifications were brought to the introduction. We agree that emphasizing the impact of aircraft NOx emissions can bring confusion as the scope is only to assess the models using the IAGOS database, but mentioning the ACACIA project and the estimation of the impact of aircraft NOx emissions remains important as it is the main motivation for this study. Thus we tried to make the introduction clearer about the aim of this study by merging the paragraph presenting ACACIA into another paragraph. Now the introduction is organized as follows:

- Importance of ozone, water vapour, CO, and $NO_x$ in the UTLS for the climate system
- Why $NO_x$ emissions are a factor controlling UTLS composition when they are emitted in the free troposphere
- Why simulating UTLS chemical composition is important to understand the impact of these high-altitude NOx emissions
- In this context, assessing the models in the UTLS is crucial
- For this purpose, the IAGOS database is well-suited
- For this use, the Interpol-IAGOS software is well-suited
- This study aims to assess 4 CCMs/CTM in the UTLS, for the ACACIA EU project

Other comments:

Line 40: How do you differentiate the impact of aviation NOx emissions from lightning NO emissions?

We generalized as *"free-tropospheric NOx emissions"* instead, thus including lightning, in order to avoid some confusion. Aviation emissions are still mentioned later in the introduction, but we hope we managed to rephrase it such that it does not seem to be the topic of this study anymore.

In the companion papers (but not in this study), the impact of aviation NOx emissions is investigated with a perturbation approach, i. e. using a couple of runs. One run includes aviation emissions, and the other one is made without it. Then we analyze the difference between the two runs.

Line 54-67: This paragraph describes the impact of individual processes on measured species but does not explain how the IAGOS dataset can assess the sensitivity of model responses to aircraft emissions.

We apologize for the confusion, we hope that our explanations (above) and clarification in the manuscript are sufficient to address this question.

Line 75: Do most measurements during the cruise occur in the LS? Are measurements in the UT primarily taken during departure and landing, potentially limiting the spatial representation of observed climatology in the UT?

In the extra-tropics, there are more measurements in the LS, especially in winter. However, the cruise measurements remain the first source of measurements in the UT. Precision has been added in the text below:

"The monitoring began in 1994 for ozone and $H_2O$, 1997 for $NO_y$, and 2001 for CO, with an abundant sampling in most of the northern extratropics (above and below the tropopause) and several tropical transects."

Table 1: It would be helpful to indicate whether these models simulate chemistry in the stratosphere.

Yes, it is a required condition for this paper. We added this information in the general comment P6, L121.

Line 175: What is meant by "online model" here? Does it imply no interaction between chemistry and meteorology?

We guess that Reviewer 2 meant "offline model". It does imply no interaction between chemistry and meteorology, or more exactly, no impact from chemistry on meteorology.

Line 283-293: How do model simulated tropopause heights compare to layers defined by PV fields? Could differences in tropopause height affect the comparison between IAGOS data and model results?

The PV fields are taken from the models' output. The model simulated tropopause and the layers are thus defined by the same data set. In order to avoid confusion, we clarified the text as follows:

"For each model, the tropopause is defined dynamically as the isosurface of 2 PVU (potential vorticity units) derived from the model output. The UT spreads from 400 hPa up to the **tropopause level** but excludes the top grid cell in order to avoid the strongest mixing zone, directly impacted by both layers (e.g. Thouret et al. (2006); Cohen et al. (2018)). The LS corresponds to all the sampled grid cells above the 3 PVU isosurface."

To the second question, we investigated the mean pressure difference between observations and modelled tropopause (2 PVU isosurface), represented in the new figures in the supplementary material (Figs. 1–5).

For a given layer, the pressure differences are relatively low across the models. Generally, it is not higher than 5 hPa, except in winter and spring in the LS, for $NO_y$ and ozone, but it does not reach 10 hPa (~ 200 m). It could still be problematic for water vapour in the LMS as the vertical gradient from the tropopause is particularly high, including two orders of magnitude (Zahn et al., 2014), but there is no apparent link between Figs. 5–8 and Figs. S2–S5. For instance, ozone (Fig. 5) is higher in the UT with EMAC and lower with LMDZ-INCA, but the sampled grid cells are closer to the model tropopause (Fig. S2) for LMDZ-INCA. In the LS, the sampled grid cells are closer to the tropopause for EMAC and UKESM, and yet, they do not minimize ozone.

The comparison with Figs. S2–S5 is added to the manuscript as follows:

*"This issue might be partly addressed by our methodology, notably the definition for the layers that enhances the isolation between them, and the exclusion of the grid cells with an inconsistent PV value regarding ozone observations. The mean pressure differences between observations and the model 2-PVU tropopause shown in Supplementary Material (Figs. S1--S5) do not exhibit noticeable differences between the models: mostly, they are less than 5 hPa (except in winter and spring for ozone and $NO_y$), and are always less than 10 hPa. For each species and layer, the distance between the sampled grid cells and the tropopause does not vary enough across the models to play a significant role in the inter-model discrepancies, as there is no apparent correlation with the chemical tracers."*

Line 327: In Section 3.1, the O3/CO ratio is used to indicate biases in cross-tropopause exchanges, while Section 3.2 attributes H2O variation to cross-tropopause mixing as well. These two sections could either be combined or require additional explanation to clarify the differences.

It can effectively seem redundant as it concerns the same thematic (cross-tropopause exchanges). The main difference is that the explanation suggested in Section 3.1 concerns biases in cross-tropopause exchanges, whereas the diagnostic for water vapour in Section 3.2 does not mention biases, only $H_2O$ differences between the UT and the LS. It only explains the lower-stratospheric behaviour of water vapour with cross-tropopause exchanges. To clarify this distinction, we added this precision:

*"The lower stratosphere shows a similar pattern, though the contrast between the summertime water vapour maximum and the rest of the year is more pronounced* **than in the UT***. This feature is consistent with the increased impact from the troposphere during this season, and the extremely steep vertical gradient in water vapour."*

This way makes it clearer that it does not concern the comparison between observations and models.

Figure 11-13: where is the figure for the comparison of CO?

This figure has been restored. As the scatterplots have moved into the Appendix, it is now labeled as Fig. B4.

---

## Author Response (AR2)

Authors' response to Editor's comments

Dear Dr. Yann Cohen,

I sent the revised version of your manuscript to one of the referees for further review. Unfortunately, this referee did not give a response although he/she had expressed a willing to further review your manuscript before.

I think that in the Author Response you have addressed the issues raised by the referees, and the revised manuscript can be accepted for publication in ACP subject to further minor revision. For example, your Article title seems not to be so concise by using only "climatologies", which cannot be easily judged what it means by a reader who does not know the IAGOS data well. The abstract should have had fewer than 250 words. I suggest that you make necessary revisions according to the Guidelines for authors at the ACP journal webpage (https://www.atmospheric-chemistry-and-physics.net/policies/guidelines_for_authors.html).

Yours sincerely,
Jianzhong Ma

Dear Editor,

We thank you for these essential comments. The title is now clarified, and the abstract has been shortened down to 250 words, as required by the journal.

1/ The title has been modified as follows:

*"Evaluation of $O_3$, $H_2O$, CO and $NO_y$ climatologies simulated by four global models in the upper troposphere–lower stratosphere with the IAGOS measurements"*

to be compared to its previous version:
*"Multi-model assessment of climatologies in the upper troposphere–lower stratosphere using the IAGOS data"*

With this new version, we clarify the fact that IAGOS are measurements, and we added the chemical species to provide context for the word "climatologies". We keep the latter to make sure the reader does not expect any time series, or trend calculation.

2/ The abstract has been reduced. The new version is quoted below:

[revised manuscript text omitted]